# Effect of gut microbiome modulation on muscle function and cognition: the PROMOTe randomised controlled trial

Mary Ni Lochlainn [1] ✉, Ruth C. E. Bowyer [1,2], Janne Marie Moll [3], María Paz García[1], Samuel Wadge[1], Andrei-Florin Baleanu[1], Ayrun Nessa[1], Alyce Sheedy[1], Gulsah Akdag[1], Deborah Hart [1], Giulia Raffaele [4], Paul T. Seed [5], Caroline Murphy[6], Stephen D. R. Harridge [7], Ailsa A. Welch [8], Carolyn Greig[9,10,11], Kevin Whelan [12] & Claire J. Steves[1] ✉

Studies suggest that inducing gut microbiota changes may alter both muscle physiology and cognitive behaviour. Gut microbiota may play a role in both anabolic resistance of older muscle, and cognition. In this placebo controlled double blinded randomised controlled trial of 36 twin pairs (72 individuals), aged ≥60, each twin pair are block randomised to receive either placebo or prebiotic daily for 12 weeks. Resistance exercise and branched chain amino acid (BCAA) supplementation is prescribed to all participants. Outcomes are physical function and cognition. The trial is carried out remotely using video visits, online questionnaires and cognitive testing, and posting of equipment and biological samples. The prebiotic supplement is well tolerated and results in a changed gut microbiome [e.g., increased relative *Bifidobacterium* abundance]. There is no significant difference between prebiotic and placebo for the primary outcome of chair rise time (β = 0.579; 95% CI −1.080-2.239 p = 0.494). The prebiotic improves cognition (factor score versus placebo (β = −0.482; 95% CI,−0.813, −0.141; p = 0.014)). Our results demonstrate that cheap and readily available gut microbiome interventions may improve cognition in our ageing population. We illustrate the feasibility of remotely delivered trials for older people, which could reduce under-representation of older people in clinical trials. ClinicalTrials.gov registration: NCT04309292.

The average age of the population is rising worldwide. As a result, the prevalence of age-related conditions and time spent living with age-related morbidity[1], including muscle loss and cognitive decline, are increasing. The prevalence of dementia is growing globally[2], but as the population ages, recognition of cognitive changes that can happen as part of healthy ageing[3] will become increasingly crucial for researchers and clinicians working with older people. Exercise can slow muscle loss and cognitive decline[4,5]. There are many reasons why older people may struggle to undertake exercise regimens, which, given their known efficacy for improving health and function, presents a challenge.

Interventions that clearly demonstrate provide a broad array of physical and mental benefits are thus needed.

Humans lose skeletal muscle with advancing age, and this can progress to sarcopenia. Dietary protein is crucial for maintaining skeletal muscle health; however, several factors can lead to reduced protein intake in older age, including social isolation, dysphagia, and slower gastric emptying[6]. In addition to consuming less dietary protein, research has shown that older adults display anabolic resistance to protein intake, a blunted responsiveness of older muscle in terms of muscle protein synthesis, compared with younger adults[7]. This has led

to a higher daily intake of 1–1.3 g/kg/day being recommended by experts[8], compared with the UK Reference Nutrient Intake for adults of 0.8 g/kg/day.

Skeletal muscle mass is regulated by the processes of muscle protein synthesis and breakdown (MPS and MPB). MPS rates are largely controlled by responsiveness to anabolic stimuli, including food consumption and physical activity. Catabolic stressors include illness, physical inactivity, and inflammation, of which older people tend to have higher rates. The aetiology of anabolic resistance is complex, involving ageing physiology and physical inactivity. Studies of protein supplementation for muscle function have displayed the most convincing results when combined with resistance exercise[8]. Resistance exercise is well established as a potent anabolic stimulus for skeletal muscle, with protein and exercise displaying a synergistic effect when used in combination[9–11].

Recent research suggests that the gut microbiome may be important for both cognitive and physical functioning in ageing. The gut microbiome is comprised of the bacteria, archaea, viruses, and eukaryotic microbes that reside in the gut, their collective genomes and the surrounding environment[12]. Its role in maintaining healthy physiology is a rapidly evolving field of enquiry. With age, the resilience of the gut microbiome is reduced, as it becomes more vulnerable to disease, medications, and lifestyle changes, with changed species richness and increased inter-individual variability[13–16]. The potential of the gut microbiota to alter physiology has been shown by landmark animal faecal transplant studies that have demonstrated body composition changes in the recipient reflective of the donor's phenotype[17]. This highlights the role of microbiota in characterising metabolic phenotypes. Several mechanisms have been proposed for anabolic resistance, and the gut microbiome has been speculated to play a role in many of these[6,18–21]. Examples include protein digestion and absorption, gut barrier function, and inflammation[6]. Further, there is evidence that the gut microbiome may influence skeletal muscle via catabolic pathways[22,23].

Prebiotics are food components that are selectively utilised by the gut microbiome to improve health[24]. Administration of a prebiotic food supplement has been shown to improve two of the Fried frailty criteria, namely hand grip strength and exhaustion[25] and overall frailty index level[26], in older adults. In addition, evidence is growing for the gut-brain axis, including preliminary evidence of a beneficial effect of prebiotic supplementation on cognition[27]. Thus, the gut microbiota may represent a malleable therapeutic target for the prevention and reversal of muscle loss with age and age-associated decline in cognition. No prior studies have investigated the effects of a supplement containing both protein and prebiotics on physical and cognitive function in older people. Genetic and environmental factors also impact physical and cognitive function, and randomising within twin pairs removes variance attributable to shared factors, enabling a more powerful study. To our knowledge, no other studies have investigated the effects of protein and prebiotic supplements in a twin population.

This study aims to assess whether the modulation of the gut microbiome using a prebiotic improves muscle function (as measured by 5× chair rise time, a marker of muscle strength) versus a placebo in a trial where all participants receive a protein (branched chain amino acids [BCAA]) and resistance exercise intervention. The study was designed to test whether the addition of gut microbiome modulation could augment established muscle function improvements from combined BCAA and exercise. The secondary outcomes were cognition, grip strength, short physical performance battery, appetite, and gut microbiome measures. The use of twins, who are matched so closely for both genetic and environmental factors, strengthened the study design.

## Results

A total of 626 individuals were assessed for eligibility, of whom 72 were successfully recruited (36 twin pairs). Figure 1 shows the flow of participants through the trial. Recruitment opened in May 2021. The date of the first visit for the first participant was 17/05/2021, and the date of the second (final) visit of the last participant was 20/12/2021. Table 1 contains the baseline characteristics of the study participants. The participants had an average age of 73 (range 63–83) and were 78% female (56/72). The arms were well matched at baseline, as expected, given the twin design. However, there was a difference in the two groups for the SNAQ appetite score. Thus, this was included in the subsequent analyses as a potential confounder. Appetite was included in the study as we know gut microbiome composition and protein intake are influenced by appetite[28].

Supplementary Table S4 displays dietary data recorded by participants at the study baseline and study end using myfood24 online software. Supplementary Table S5 displays the same data with over-reporters removed from the analysis. No under-reporters were identified. Supplementary Table S6 presents the dietary data by study arm. There were no differences between prebiotic and placebo groups at baseline or at the study endpoint for any measures of fibre intake. Overall, there was a small reduction in energy intake between the baseline and study endpoint, which appears to have been driven by a reduction in the prebiotic group (−132.4 kcal/d). This small difference between the prebiotic and placebo groups (75.6 kcal/d) falls within the limits of error for measurement of energy intake. The effect of this small energy intake reduction compounding over time, potentially mediated via the effect of inulin on appetite[29,30], could contribute to impacts on muscle strength. However, we are cognizant of the limitations of being able to measure such small changes in energy intake even when using gold standard dietary recording techniques in free-living individuals. There were no significant differences in body weight (kg) or body mass index between baseline and study end in either arm (Supplementary Table S6).

Supplementary Table S7 displays the bowel habit questionnaire results, with no differences found between intervention groups for any of the questions relating to bowel habits. Supplementary Table S1 displays the demographic characteristics of those who were ineligible or declined to take part versus those who took part, using existing TwinsUK longitudinal data. The cohort is majority female for historical reasons. Participants were younger and there were more males than the wider TwinsUK cohort. A significant difference was retained after adjusting the sex difference for age and age difference for sex (Supplementary Table S1).

### Adverse events and compliance

There was an excess of mild adverse events (such as abdominal bloating) in the prebiotic arm than the placebo arm, but no difference in adherence, suggesting the supplements overall were well tolerated. No participants reported 'gastrointestinal side effect' as a reason for noncompliance with the study intervention in either arm. No serious adverse events took place.

Table 2 summarises the number of adverse events per arm and compliance with the intervention. Supplementary Table S2 lists all adverse events reported by participants. We also recorded medication changes (Table 2) during the study, of which there were relatively few.

### Primary and secondary outcomes

The results of the linear mixed models for the primary and secondary outcomes are shown in Table 3, including both intention to treat (ITT) and per protocol (PP) analyses. There were no notable differences in results between the ITT and PP models. There was no difference between prebiotic and placebo for the primary outcome of chair rise time (mean change between baseline and study end 0.88 s [SD 2.16] for prebiotic and 1.12 s [SD 1.52] for placebo; $p = 0.631$), nor for the physical performance secondary outcomes (grip strength, short physical performance battery, and IPAQ MET minutes) or SNAQ appetite score.

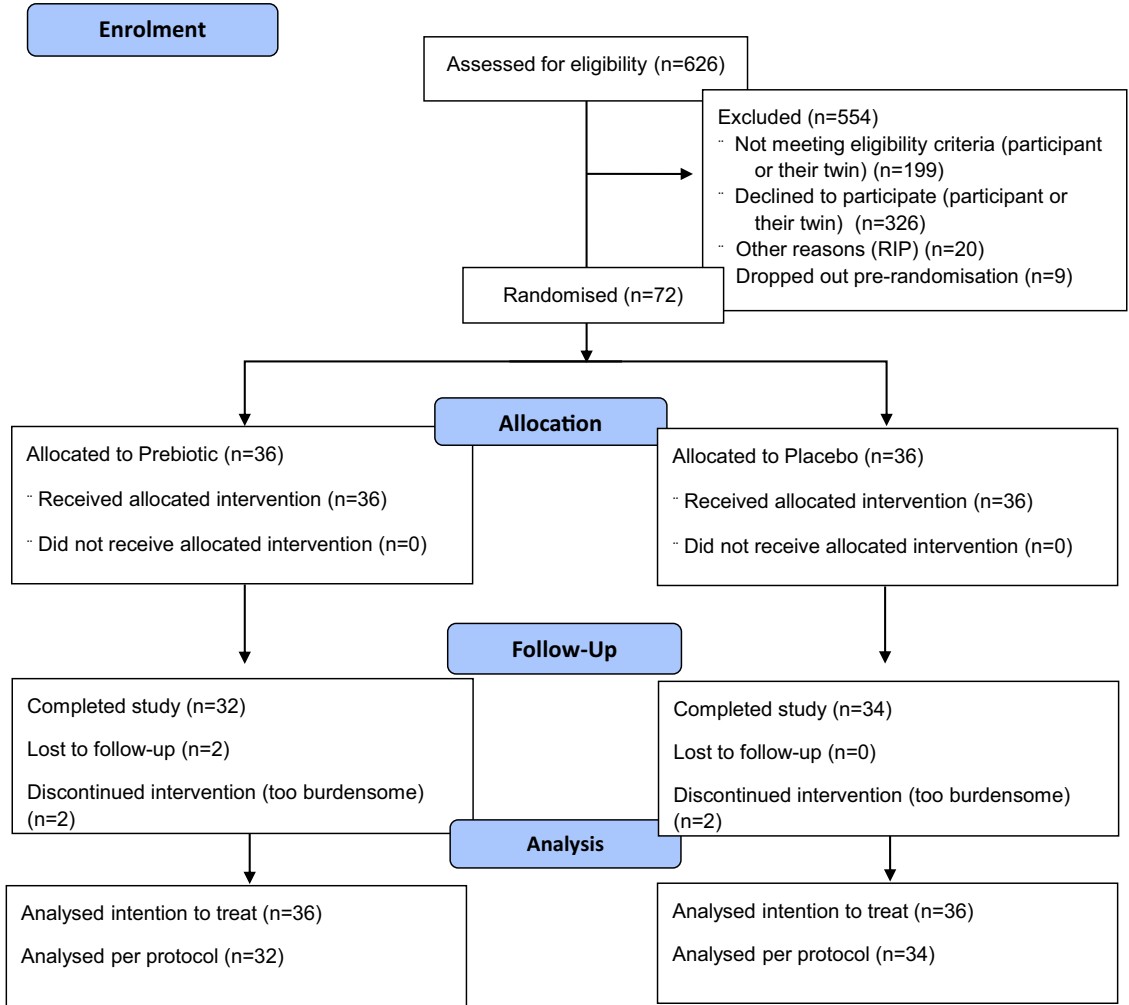

**Fig. 1 | Participant flow diagram.** Flow of participants through the trial, including assessment for eligibility, randomisation, follow-up and analysis.

The prebiotic group showed a significant improvement in the cognitive first-factor score compared to placebo ($\beta = -0.482$; 95% CI $-0.813$, $-0.141$; $p = 0.014$). In addition, the specific cognitive test paired associates learning (a memory test) had a significantly lower number of errors in the prebiotic group compared to the placebo ($\beta = 7.55$; 95% CI: 4.65–10.46; $p = 0.001$). This particular cognitive test has clinical significance, having been identified as an early identifier of Alzheimer's dementia[31,32]. Missingness of data was higher for CANTAB cognitive tests compared to other outcome variables. No significant differences were found when comparing the demographic features of those who completed cognitive tests versus those who did not, thus no specific features predicted the missingness of the cognitive tests, as shown in Supplementary Table S3. No unplanned post-hoc analyses were carried out.

**Gut microbiome results**

In addition to the secondary physical performance outcomes, gut microbiome composition and functionality were analysed and obtained from stool samples. Good sequencing data were obtained from all samples, with an average of 21.6 M read pairs per sample and a minimum of 11.7 M read pairs. (Supplementary Fig. S1). On average 17.9 M read pairs per sample could be mapped to the gene catalogue, representing on average 82.8% of the HQNH reads (min = 68.5%, see Supplementary Table S8), representing 1494 MGS. Most samples were dominated by families *Bacteroidaceae*, *Oscillospiraceae*, and *Lachnospiraceae*, with some inter-individual variation, as would be expected.

**Twin-pair microbiota similarity.** The similarity between the study's twin pairs' microbiota composition at baseline and end of the study was investigated, and a heritability analysis was conducted (Supplementary Fig. S2). Whilst there was evidence of intra-twin pair differences in microbiota composition (Supplementary Fig. S3), we found that twin pairs' microbiota were significantly more similar compared to the microbiota composition of unrelated individuals both at baseline ($U = 55579$, $p = 0.0026$) and end of study ($U = 44342$, $p = 0.00072$), using Bray–Curtis dissimilarity (Supplementary Fig. S4).

Although the p-values of the Mann–Whitney $U$ test comparing twin pairs to non-twin pairs are significant at both timepoints, an additional statistical analysis was performed to verify that these findings were not due to the high number of pairwise observations in the non-twin group. For a comparison of 35 twin pairs, there are many more observations comparing two non-twins ($35 * (35 - 1)/2 = 595$ observations) than two twins (35 observations), which impairs the validity of a simple statistical two-group comparison due to high number of observations in the non-twin group. Therefore, a permutational approach was used to assess whether the obtained test statistic for the group comparison was significantly different from the test statistic distribution obtained by randomly permuting the labels of the observations (Supplementary Fig. S5). For both baseline and study end, the *p*-value of comparisons between the true test statistic and the test statistic distribution obtained from permutations was practically zero (Mann–Whitney $U$ test, $P = 3.33 * 10^{-165}$ and $3.33 * 10^{-165}$, for baseline and end of study, respectively), indicating that microbiota

**Table 1 | Baseline Characteristics, by study arm**

| | | Total N = 72 | Prebiotic N = 36 | Placebo N = 36 | p-Value |
|---|---|---|---|---|---|
| Age (years) | | 73.1 (4.8) | 73.1 (4.9) | 73.1 (4.9) | 1.000 |
| Sex | Female | 56 (77.8%) | 28 (77.8%) | 28 (77.8%) | 1.000 |
| | Male | 16 (22.2%) | 8 (22.2%) | 8 (22.2%) | |
| Zygosity | Monozygotic | 44 (61.1%) | 22 (61.1%) | 22 (61.1%) | 1.000 |
| | Dizygotic | 28 (38.9%) | 14 (38.9%) | 14 (38.9%) | |
| Smoking Status | Never Smoked | 38 (52.8%) | 19 (52.8%) | 19 (52.8%) | 0.570 |
| | Ex-Smoker | 30 (41.7%) | 14 (38.9%) | 16 (44.4%) | |
| | Current Smoker | 4 (5.6%) | 3 (8.3%) | 1 (2.8%) | |
| Household Income | Declined to Answer | 13 (18.1%) | 5 (13.9%) | 8 (22.2%) | 0.590 |
| | Low Income | 30 (41.7%) | 14 (38.9%) | 16 (44.4%) | |
| | Middle Income | 23 (31.9%) | 14 (38.9%) | 9 (25.0%) | |
| | High Income | 6 (8.3%) | 3 (8.3%) | 3 (8.3%) | |
| Education Level | Low | 33 (45.8%) | 18 (50.0%) | 15 (41.7%) | 0.730 |
| | Middle | 25 (34.7%) | 11 (30.6%) | 14 (38.9%) | |
| | High | 14 (19.4%) | 7 (19.4%) | 7 (19.4%) | |
| Alcohol Intake | Never Drink | 20 (27.8%) | 12 (33.3%) | 8 (22.2%) | 0.510 |
| | Less than weekly | 17 (23.6%) | 7 (19.4%) | 10 (27.8%) | |
| | At least weekly | 35 (48.6%) | 17 (47.2%) | 18 (50.0%) | |
| Weight (kg) | | 73.8 (15.7) | 71.8 (14.0) | 75.8 (17.2) | 0.280 |
| Height (m) | | 1.6 (0.1) | 1.6 (0.1) | 1.6 (0.1) | 0.950 |
| Body Mass Index | | 28.0 (5.1) | 27.3 (5.0) | 28.7 (5.3) | 0.240 |
| Protein Intake (g/d) | | 60.7 (14.4) | 60.1 (16.2) | 61.3 (12.5) | 0.710 |
| Protein/Weight (g/kg/d) | | 0.9 (0.2) | 0.9 (0.2) | 0.8 (0.2) | 0.910 |
| Energy Intake (kcals/d) | | 1569.5 (397.7) | 1581.6 (380.5) | 1557.5 (419.2) | 0.800 |
| Frailty Index | | 0.2 (0.1) | 0.2 (0.1) | 0.2 (0.1) | 0.790 |
| Gait speed (m/s) | | 1.1 (0.4) | 1.1 (0.4) | 1.1 (0.3) | 0.640 |
| Chair Rise Time (s) | | 10.6 (3.3) | 10.3 (3.2) | 11.0 (3.5) | 0.370 |
| Grip Strength (kg) | | 26.0 (9.7) | 25.7 (9.5) | 26.3 (10.0) | 0.800 |
| Balance Score | | 4.0 (0.2) | 4.0 (0.0) | 3.9 (0.2) | 0.160 |
| SPPB Score | | 11.0 (1.5) | 10.9 (1.5) | 11.0 (1.5) | 0.810 |
| IPAQ MET mins/week | | 674.8 (654.5) | 564.8 (591.2) | 784.8 (703.2) | 0.160 |
| IPAQ Score | Low activity | 69 (95.8%) | 35 (97.2%) | 34 (94.4%) | 0.560 |
| | Medium activity | 3 (4.2%) | 1 (2.8%) | 2 (5.6%) | |
| Appetite—SNAQ | | 15.2 (2.0) | 14.7 (2.0) | 15.8 (1.9) | 0.046* |
| Cognition (CANTAB) | | | | | |
| Executive function | OTS—mean latency (speed of response; ms) | 32773.2 (20028.7) | 37476.2 (24711.6) | 28766.9 (14250.0) | 0.130 |
| | Spatial working memory (no. errors) | 14.3 (10.0) | 13.0 (8.7) | 15.3 (11.0) | 0.420 |
| Memory | PAL: first attempt memory score | 11.9 (4.6) | 12.5 (4.5) | 11.4 (4.7) | 0.430 |
| | PAL: total no. errors | 18.0 (16.2) | 16.7 (15.9) | 19.2 (16.7) | 0.580 |
| | Pattern recognition memory (% correct) | 89.5 (10.5) | 87.0 (11.7) | 91.7 (9.0) | 0.110 |
| | Spatial span (forward span length; range 2-9) | 5.7 (1.3) | 5.9 (1.6) | 5.5 (1.0) | 0.350 |
| | Missing cognitive data | 22 | 13 | 9 | |

Continuous data is presented as mean (standard deviation). Categorical data are presented as n(%). * Denotes significance. Missing data numbers are listed for cognition; no other variables have missing data. P values are group comparisons between each study arm performed using a two-sided paired t-test.
*SPPB* short physical performance battery, *IPAQ* international physical activity questionnaire, *MET* metabolic equivalent of task (One MET is the energy you expend at rest), *SNAQ* simplified nutritional assessment questionnaire, *CANTAB* Cambridge neuropsychological test automated battery, *OTS* one touch stockings of Cambridge cognitive test, *PAL* paired associates learning.

composition is significantly more similar within twin pairs than between two individuals from different twin pairs.

**Microbiota between prebiotic and placebo.** No microbiota features were significantly different between the prebiotic and placebo at baseline as assessed by PERMANOVA. Figure 2 displays the comparisons between groups at the study endpoint, revealing 11 significant differences, mostly driven by a higher relative abundance of *Bifidobacterium* in the prebiotic group, including higher abundance from phylum level of Actinobacteria down to the genus level, smaller increases in the phyla Firmicutes and Bacteroidetes, and a decrease in the species *Phocea massiliensis* and its cognate genus. In addition, Fig. 2 displays the fifteen microbiota features found to be significantly different between the prebiotic and the placebo groups when

**Table 2 | PROMOTe adverse events, medication changes, compliance, and dropouts by study arm**

| | Total N = 72 | Prebiotic arm N = 36 | Placebo arm N = 36 | p-Value |
|---|---|---|---|---|
| Dropped out of study n(%) | 4 (6%) | 2 (6%) | 2 (6%) | 1 |
| Adverse event n(%) | 10 (7%) | 8 (22%) | 2 (6%) | 0.041* |
| Medication change n/N(%) | 7/60 (12%) | 2/29 (7%) | 5/31 (16%) | 0.270 |
| *Compliance* | | | | |
| Exercises n/N(%) | 59/59 (100%) | 30/30 (100%) | 28/29 (97%) | – |
| Sachets n/N(%) | 59/59 (100%) | 30/30 (100%) | 29/29 (100%) | – |
| Sachet count, % adherence mean (SD) | 78.7% (9.6) | 79.7% (10.5) | 77.5% (8.6) | 0.370 |

Data presented as n(%). For data with incomplete data available, these data are presented as n/N(%). P values are group comparisons between each study arm performed using a two-sided paired t-test. *indicates statistical significance.

comparing relative abundances at the study end adjusted for baseline abundances and the twin pair. In addition to the main effects of increased Bifidobacterium and decreased *P. massiliensis* observed for the cross-sectional analysis, the prebiotic group has a lower relative abundance of *Anaeromassilibacillus* and a selection of higher-level taxonomies, such as Deltaproteobacteria, Lactobacillales, and Eubacteriales.

There was no significant difference between prebiotic and placebo groups for any of the alpha diversity measures. There were also no significant differences in beta diversity between prebiotic and placebo groups at the study end as assessed by PERMANOVA. Intra-pair dissimilarity for beta diversity was not found to be higher at the study end than at the study baseline ($V = 102$; $p = 0.58$). Supplementary Fig. S7 displays the comparison of Bray–Curtis dissimilarity between twin pairs, including subgrouping by zygosity.

When undertaking within-group comparisons between baseline and study end, there were 40 significantly different features when comparing samples from the prebiotic arm while only the relative abundance of *Actinomyces graevenitzii* was significantly different when comparing samples from the placebo arm (Supplementary Fig. S6).

**Microbiota features and muscle strength.** In a compositional bias-corrected linear model adjusting for arm- and twin pair-related effects, 129 microbiota features were significantly correlated with chair rise time, including 95 microbial taxa, 32 microbial functions, and richness and Faith's phylogenetic diversity (Supplementary Fig. S8). Correlation analysis (Pearson's correlation coefficient from models using centred log-ratio transformed abundances) between change in chair rise time and change in microbiota features over the study intervention period (analysing all participants together), revealed a correlation between improvement in chair rise time with both change in richness ($r = -0.347$; $p_{adj} = 0.0159$), Shannon diversity ($r = -0.250$; $p_{adj} = 0.0486$), and Faith's diversity ($r = -0.297$; $p_{adj} = 0.0275$) (Supplementary Fig. S9). The p-values presented here have been adjusted for multiple testing.

**Microbiota features and cognition.** Using the approach for the analysis of the correlation between physical ability (chair rise time) and microbiota features above, we observed five microbiota features that correlated significantly with cognitive ability (cognition factor score) when adjusting for arm- and twin pair-related effects and bias-correction for compositional effects of taxon abundances (Supplementary Fig. S10). This included the phylum Actinobacteria

($r = -0.323$; $p_{adj} = 0.0447$), which was significantly increased in the prebiotic group compared to the placebo at study end. Correlations between improvement in cognition factor score were also associated with increases in relative abundance of Veillonellaceae and its cognate order Veillonelalles from baseline to study end ($r = 0.585$, $p_{adj} = 0.0555$, and $r = 0.543$, $p_{adj} = 0.0772$, respectively (Supplementary Fig. S9).

## Discussion

Ageing is associated with increased frailty and worsening of cognition, both of which may involve the role of the gut microbiome. In this RCT, we demonstrated that a prebiotic was generally well tolerated and resulted in a changed gut microbiome, in particular with the increased relative abundance of *Bifidobacterium*. Despite this, there was no significant difference in the primary outcome of chair rise time between the prebiotic and placebo, however prebiotic improved cognition compared with placebo.

### Muscle strength and function

No evidence was found that this prebiotic improved muscle strength compared with placebo in a 12-week time frame, in particular with our primary outcome of chair rise time. In addition, there was no effect of the prebiotic on any of the other secondary outcomes related to muscle strength and function including hand grip strength, SPPB, or physical activity (IPAQ).

The 12-week intervention period was chosen based on previous prebiotic intervention studies[27,33]. However, none were focused on muscle strength as their primary outcome. This timeframe may have been insufficient for muscle remodelling to take place, and it is possible that a longer intervention period may be needed to appreciate the influence of gut microbiome modulation on muscle health. While the EWGSOP2 guidance recommends chair rise time as a muscle strength measure, ensuing work should consider adding an isometric measure of muscle strength which has the ability to detect subtle changes in the muscle (e.g., quadriceps) in addition to chair rise time. While considered highly functionally relevant for older people, chair rise time is a composite, relying on power, balance, vision, etc. and a proxy rather than a direct measure of strength. Wide variability has been reported in muscle strength measures in older people[34], although measures such as chair rise time and grip strength are easily implemented in a clinical setting, more detailed and direct strength measures should be considered in future research aiming to prove the efficacy of gut microbiome modulation in improving muscle outcomes for older adults.

Furthermore, our sample size calculation was based on previous trials that used chair rise time as their primary outcome[35–37]. However, no trial existed that had used a microbiome intervention, and therefore it is possible that the study was underpowered. Indeed, only one-fifth (14/65; 8 in the prebiotic arm, 6 in the placebo arm, $p = 0.68$) of those who had chair-rise time measured at both baseline and study endpoint achieved a 20% improvement, which was the estimated improvement used in the sample size calculation.

The dose of prebiotic was based on a trial using the same intervention in nursing home residents[25]. A study by Tandon et al. (2019) investigated fructooligosaccharide (FOS) dosing in younger adults and reported no statistically significant differences in gut microbiota changes in response to 2.5 g/d, 5 g/d and 10 g/d doses (PROMOTe sachets contained 7.5 g/d), however there were trends towards differences in the abundances of *Lactobacillus* in the highest dosing group[38]. So et al. carried out a meta-analysis of dietary fibre interventions, many of which included inulin and/or FOS, and noted no differences in effect on *Bifidobacterium* abundance with varying doses[39]. They hypothesised that there may be a limit to the amount of fibre that Bifidobacterium can ferment, or perhaps the lack of a dose-response effect may be attributable to the percentage increase from baseline

**Table 3 | Linear mixed-effects model results for primary and secondary outcomes**

| Outcome | | Prebiotic | Placebo | Treatment effect (prebiotic = reference group) | | | | | |
|---|---|---|---|---|---|---|---|---|---|
| | | | | Intention to treat | | | Per protocol | | |
| | | | | Coefficient | 95% CI | *P*-value | Coefficient | 95% CI | *P*-value |
| *Primary outcome* | | | | | | | | | |
| Chair rise time (secs) | | | | 0.58 | 1.08,2.24 | 0.494 | 0.58 | 1.08,2.24 | 0.494 |
| Baseline | | 10.38 (3.21) | 10.24 (2.21) | | | | | | |
| Study End | | 9.49 (2.89) | 9.12 (1.96) | | | | | | |
| Change between baseline and study end | | 0.88 (2.16) | 1.12 (1.52) | | | | | | |
| *Secondary outcomes* | | | | | | | | | |
| Grip strength (kg) | | | | 1.22 | −2.43,4.86 | 0.512 | 1.22 | −2.43,4.86 | 0.512 |
| Baseline | | 26.23 (9.33) | 25.52 (9.27) | | | | | | |
| Study End | | 28.98 (11.64) | 29.14 (10.08) | | | | | | |
| Change between baseline and study end | | 2.74 (10.76) | 3.45 (8.48) | | | | | | |
| SPPB score (range 0-12) | | | | −0.13 | −0.56,0.30 | 0.551 | −0.13 | −0.56,0.30 | 0.551 |
| Baseline | | 10.88 (1.49) | 11.32 (1.05) | | | | | | |
| Study End | | 11.12 (1.32) | 11.45 (0.81) | | | | | | |
| Change between baseline and study end | | 0.24 (1.05) | 0.06 (1.01) | | | | | | |
| IPAQ MET minutes | | | | 194.43 | −111.39, 500.24 | 0.213 | 245.18 | −70.36, 560.73 | 0.128 |
| Baseline | | 538.09 (596.87) | 769.30 (640.08) | | | | | | |
| Study End | | 771.09 (594.28) | 1030.83 (648.33) | | | | | | |
| Change between baseline and study end | | 256.06 (853.84) | 72.04 (866.41) | | | | | | |
| Appetite (SNAQ) (range 4-20) | | | | 0.19 | −0.54,0.92 | 0.607 | 0.24 | −0.51,0.99 | 0.532 |
| Baseline | | 14.79 (1.79) | 15.89 (1.76) | | | | | | |
| Study End | | 14.71 (1.88) | 15.80 (2.00) | | | | | | |
| Change between baseline and study end | | −0.01 (1.29) | −0.01 (1.62) | | | | | | |
| Cognition (CANTAB) | Cognition factor score | | | −0.48 | −0.81, −0.14 | 0.014* | −0.48 | −0.81, −0.14 | 0.014* |
| | Baseline | 0.05 (1.00) | −0.23 (0.94) | | | | | | |
| | Study End | 0.59 (0.68) | 0.18 (0.60) | | | | | | |
| | Change between baseline and study end | 0.40 (0.83) | 0.19 (0.70) | | | | | | |
| Executive function | OTS—mean latency (speed of response; ms) | | | 631.22 | −8910.33, 10171.76 | 0.889 | 631.22 | −8910.33, 10171.76 | 0.889 |
| | Baseline | 37,837.29 (25230.98) | 28,318.90 (14707.45) | | | | | | |
| | Study End | 32,737.31 (21417.24) | 26,555.30 (13306.24) | | | | | | |
| | Change between baseline and study end | −1.0e + 04 (13,978.77) | −6594.90 (9405.05) | | | | | | |
| | Spatial working memory (no. errors) | | | −1.84 | −7.12,3.43 | 0.457 | −1.84 | −7.12,3.43 | 0.457 |
| | Baseline | 12.77 (8.89) | 15.13 (11.52) | | | | | | |
| | Study End | 10.15 (6.39) | 11.23 (9.16) | | | | | | |
| | Change between baseline and study end | −3.08 (9.22) | −5.46 (10.44) | | | | | | |
| Memory | PAL: first attempt memory score | | | −1.95 | −4.43,0.52 | 0.104 | −1.95 | −4.43,0.52 | 0.104 |
| | Baseline | 12.41 (4.64) | 10.88 (4.44) | | | | | | |
| | Study End | 14.54 (3.99) | 12.69 (3.38) | | | | | | |
| | Change between baseline and study end | 1.77 (4.53) | 0.77 (4.19) | | | | | | |
| | PAL: total no. errors | | | 7.55 | 4.65, 10.46 | 0.001* | 7.55 | 4.65, 10.46 | 0.001* |
| | Baseline | 17.05 (16.13) | 20.67 (17.05) | | | | | | |
| | Study End | 9.15 (7.05) | 12.23 (10.15) | | | | | | |
| | Change between baseline and study end | −5.15 (13.92) | −5.00 (12.34) | | | | | | |
| | Pattern recognition memory (% correct) | | | −3.00 | −9.40,3.40 | 0.331 | −3.00 | −9.40,3.40 | 0.331 |
| | Baseline | 86.74 (11.97) | 91.32 (9.35) | | | | | | |
| | Study End | 97.44 (7.12) | 93.59 (7.72) | | | | | | |

**Table 3 (continued) | Linear mixed-effects model results for primary and secondary outcomes**

| Outcome | | Prebiotic | Placebo | Treatment effect (prebiotic = reference group) | | | | | |
|---|---|---|---|---|---|---|---|---|---|
| | | | | Intention to treat | | | Per protocol | | |
| | Change between baseline and study end | 8.33 (7.61) | 1.92 (9.71) | | | | | | |
| | Spatial span (forward span length; range 2–9) | | | −0.86 | −1.97,0.26 | 0.120 | −0.86 | −1.97,0.26 | 0.120 |
| | Baseline | 5.91 (1.63) | 5.50 (1.06) | | | | | | |
| | Study End | 6.69 (1.60) | 5.46 (0.66) | | | | | | |
| | Change between baseline and study end | 0.54 (1.56) | −0.23 (1.09) | | | | | | |

Prebiotic group = reference group * Denotes significance.
Continuous data are presented as mean (standard deviation), and categorical are presented as *n* (%). All linear mixed models are adjusted for SNAQ score at baseline.
*CI* confidence interval, *SPPB* short physical performance battery, *IPAQ* international physical activity questionnaire, *MET* metabolic equivalent of task (One MET is the energy you expend at rest), *SNAQ* simplified nutritional assessment questionnaire, *CANTAB* Cambridge neuropsychological test automated battery, *OTS* one touch stockings of Cambridge cognitive test, *PAL* paired associates learning.

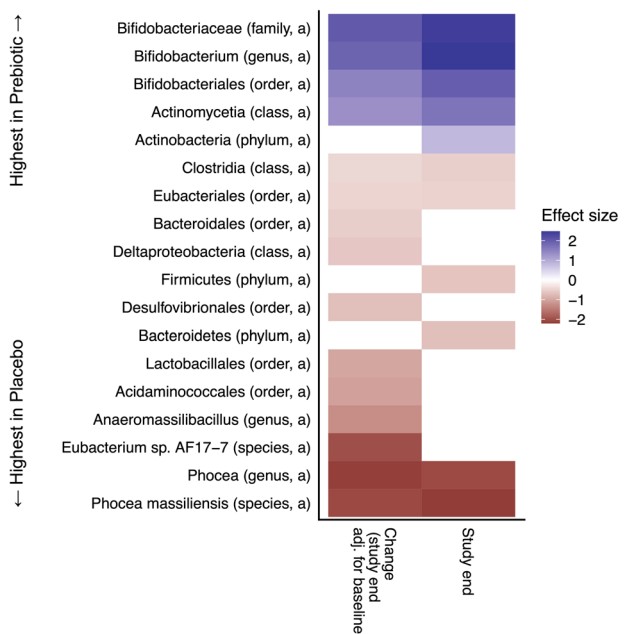

**Fig. 2 | Twin-paired group comparisons between prebiotic and placebo supplements, including data from the end of the study or the difference between baseline and end of study.** Paired group comparison of relative abundance (a) and prevalence (p) of bacterial taxa between study arms cross-sectionally or at study end adjusted for baseline taxon abundances. Testing for differences in microbiome taxon abundances was performed with a linear regression framework with a compositional bias correction based on LinDA. There were no significant differences between groups at baseline or for gut metabolic modules, gut–brain modules, or alpha diversity. Effect sizes are the bias-corrected coefficients from the linear models where positive effect sizes (blue) indicate higher levels in the placebo group and negative effect sizes (red) indicate higher levels in the prebiotic group.

rather than the intervention dose[39]. The findings presented here suggest the 7.5 g/d dose was sufficient to elicit changes within the gut microbiome. However, no literature exists on sufficient dosing of prebiotics in terms of physical outcomes such as muscle strength. It is possible that the dose, while sufficient to elicit changes within the gut microbiome, was insufficient to provide a clinically important response in terms of muscle strength within a healthy volunteer cohort.

Lastly, while the prebiotic used in this trial was shown to improve grip strength, exhaustion levels, and frailty index in another trial of older participants[25,26], it is possible the inulin/FOS prebiotic is not the

optimal prebiotic for influencing muscle health, and further research is needed to establish the best prebiotic for this aim.

### Cognition
In terms of secondary outcomes, the prebiotic improved cognition compared with placebo. By using a factor score to combine the components of the CANTAB cognitive battery, it is difficult to report whether this improvement is clinically meaningful. The only individual cognitive test which had a significant improvement in the prebiotic versus placebo groups was the number of errors in the Paired Associates Learning test, which is a memory test particularly focused on visual memory and new learning[40]. This measure has also been identified as an early marker of disease phenomena such as Alzheimer's Disease[31,32].

The concept of a gut–brain axis, with the microbiota influencing bidirectional gut-brain pathways, has become established in recent years[41,42]. There are thought to be three main pathways for this bidirectional communication: neural, immune, and endocrine[41,43,44]. Indeed, there is a growing body of evidence supporting interventions which target the gut microbiome (including prebiotic) in improving cognition[27]. Desmedt et al. reviewed intervention studies of prebiotics on human affect and cognition and found 14 studies with wide variations in methodologies, outcomes, prebiotic dose, and intervention duration[27]. Most of the studies focused on mood and affect but there were improvements seen in verbal episodic memory and recognition reported. None of the studies used the CANTAB cognitive tests. Thus, to confirm our findings, a larger study which includes a range of cognitive tests, alongside measures of effect and attention, and detailed microbiome measures (compositional and functional) is warranted.

### Gut microbiome
Importantly, the prebiotic intervention impacted the gut microbiota composition and gut microbiota function. Forty microbiome features changed between baseline and study end for the prebiotic group, while only one was changed in placebo (Supplementary Fig. S6). These features include alpha diversity and relative abundance of bacterial taxa. A reduction in alpha diversity was observed in the prebiotic arm, which is in keeping with the findings of other research groups[39,45,46]. Prebiotics tend to cause an increased abundance of certain taxa rather than greater diversity, as seen in Fig. 2 and Supplementary Fig. S6, which logically can cause a reduction in diversity. The prebiotic group had a significant increase in the relative abundance of Actinobacteria, particularly *Bifidobacterium*, compared to placebo, which has been found elsewhere with the administration of FOS and/or inulin prebiotic[39,46–50]. Notably, no increase in *Lactobacillus* or *Faecalibacterium* was seen, both of which have been reported by numerous others in studies of prebiotic supplementation, in particular, inulin-

type fructans[38,47–49,51], while the relative abundance of the two genera's cognate orders, Lactobacillales and Eubacteriales, respectively, were increased in the prebiotic group compared to placebo, indicating that closely related genera could play a role as well.

Interestingly, despite one twin from each pair receiving prebiotic and the other placebo, there was no significant change in intra-twin dissimilarity at the study endpoint. Previous work has shown that the gut microbiome is shaped by genetics and early environment[52–55], with early seeding and succession of the community ecology. Thus, the 12-week prebiotic intervention was likely insufficient to change the similarity between twin pairs, who share genetics, early environmental exposures, and often many other factors. The heritability analysis revealed that some of the taxa which we found to have changed over the course of the intervention (e.g. *Bifidobacterium*) were heritable, which indicates that our study design was strengthened by using twin pairs, and the design is likely to have increased our ability to detect an effect.

Moving from composition to *function* of the gut microbiota, several GMMs were found to be significantly correlated to physical ability independently of study arm and twin pairing (Supplementary Fig. S8). Among these were metabolic pathways for amino acids and carbohydrates, as well as the pentose phosphate pathway; the pathway provides precursors for amino acid biosynthesis and intermediates for anabolism[56] and is crucial in skeletal muscle anabolism[57,58]. This finding adds weight to the gut-muscle axis hypothesis, which argues that a bidirectional link exists between the gut microbiota and skeletal muscle, which may have mechanistic implications in age-associated muscle atrophy[18], and warrants further investigation in future studies focused on the role of the gut microbiome in age-associated anabolic resistance of skeletal muscle.

There is a growing body of evidence supporting a gut-brain axis, a bidirectional communication system between central and enteric nervous systems, in which the gut microbiota plays a key influential role[59]. Here, we report a positive correlation between increases in the cognitive factor score and the relative abundance of Actinobacteria at the study end, even after adjusting for baseline abundance, study arm, and twin pairing (Supplementary Fig. S10). Actinobacteria has been associated with cognition elsewhere and has been implicated in Alzheimer's disease[60,61] and Parkinson's disease[62]. The direction of association varies among studies. While it has been associated with disease states, one study of 39 adults reported higher abundance being associated with improved motor speed and attention[63]. Actinobacteria is the phylum to which the genera *Bifidobacterium* belongs; thus, it is noteworthy that several studies of *Bifidobacterium*-based probiotic interventions have shown promising improvements in cognition[64]. More research is needed to understand the connection between Actinobacteria and cognitive function.

## Study feasibility

The remote nature of this clinical trial is a clear strength and proves that remotely delivered interventions can be successfully administered in older adults, with substantial data collected and good adherence. This study design holds promise for improving the under-representation of older adults in research. National bodies have called for greater inclusivity in clinical research, identifying older adults as an under-served group, and acknowledged travel as a key limiting factor in their participation[65,66]. In addition, the exclusion criteria for recruitment were designed to be as inclusive as possible to older people, including those living with frailty and multi-morbidity.

More adverse events were noted in the prebiotic group; however, these were largely expected side effects, which participants had been informed of a priori. There was no difference in adherence between the two arms, suggesting that these side effects did not affect compliance. Thus, this study has shown the feasibility of administering a relatively cheap and commercially available gut microbiome intervention to a population of older adults. Future larger-scale studies using larger

sample sizes, longer intervention periods, and possibly different dose profiles could examine the potential of the gut microbiome to mediate anabolic resistance of skeletal muscle to protein in older adults.

## Strengths and limitations

A key strength of this trial is the use of twin pairs randomised to each study arm, which resulted in very closely matched arms at baseline, improving the ability to detect meaningful effects and reducing confounding. In addition, the design was a pragmatic trial-within-cohort, which led to successful recruitment of the sample size to time and target.

The remote study design is a strength, as there is a clear need for innovative trial designs in a post-COVID research landscape, with reduced need for travel for older participants, who may not wish to come to a hospital setting or who may have mobility impairments. Remote trials may also lead to reduced research costs. However, remote delivery can exclude those who are not digitally literate or who do not have access to a computer/tablet and/or the internet. This may disproportionately affect those with lower education, greater deprivation levels, English as a second language, and other related factors[66]. Data collection may also be less precise than when performed in person, although the use of video teleconferencing reduced this problem.

The remote design led to an inability to measure muscle mass, which is a limitation. While muscle mass is one of the diagnostic features of sarcopenia, the European Working Group for Sarcopenia of Older Persons 2 (EWGSOP2) guidance does recommend muscle strength as the most important identifying factor[67]. Future studies investigating interventions that target the gut-muscle axis should include a muscle mass measure to examine this relationship.

Missingness of data was higher for CANTAB cognitive tests compared to other outcome variables. This was likely due to the burden of doing CANTAB at home, which can be time-consuming, and the software is not compatible with all tablet devices. Some participants who were using tablets could not get the software to work on their devices.

Gut microbiome measures were taken at two timepoints, baseline and study end. Without a mid-study sample, we are unable to comment on whether or not the microbiota findings at the study endpoint reflect a partial recovery from the prebiotic intervention. A greater sample size and longer intervention time would likely be needed to test this. Further, functional analysis of the gut microbiome in this study is based on metagenomic data rather than direct measurement. Future research could examine meta transcriptomics or metabolomics of stool to examine this further. Maltodextrin is a commonly used placebo in trials of prebiotics however it has limitations, as it is not directly energy-matched with the prebiotic. Future trials could use an energy-matched placebo to overcome this.

The TwinsUK cohort, whence participants were recruited, is majority female for historical reasons and has a healthy volunteer bias[68]. However, it is largely representative of the UK population[68]. This study aimed to look at those over 60s, and therefore, is not generalisable to younger age groups. In addition, the majority female nature of the participants somewhat limits generalisability. Future work in this area should include middle-aged adults, the oldest old, and a greater representation of men.

## Discussion summary

Prebiotics improved cognition but did not impact muscle strength and function, compared with placebo in a cohort of healthy older twins. Our results demonstrate that cheap and readily available gut microbiome interventions hold promise for improving cognitive frailty in our ageing population. While this trial did not demonstrate improvement in skeletal muscle strength, we showed that gut microbiome modulation via prebiotic supplementation in the context of ageing-muscle research is feasible and well tolerated, with clear responses noted in the gut microbiota composition and function. Future larger trials can examine the use of gut microbiome targeting interventions to overcome age-associated

| Stage | 1 | 2 | 3 | 4 | 5 | 6 | 7 |
|---|---|---|---|---|---|---|---|
| Assessment | Eligibility | Pre-baseline video call | Baseline video call (week 1) | Week 3-5 | Week 7-9 | Pre-final call | Final video call (week 12) |
| Inclusion/Exclusion Criteria checks | X | | | | | | |
| Participant information and informed consent | X | | X | | | | |
| Randomisation | | | X | | | | |
| Postal box sent out | | X | | | | | |
| 3-day food diary (online) | | X | | | | X | |
| Questionnaire (online) | | X | | | | X | |
| CANTAB cognitive test (online) | | X | | | | X | |
| Provide supplements (post) | | | X | | | | |
| Stool sample (post) | | | X | | | X | |
| Short Physical Performance Battery | | | X | | | | X |
| Weight (kg) | | | X | | | | X |
| Height (cm) | | | X | | | | X |
| Compliance/adverse effects checks | | | | X | X | | |
| Count leftover supplement sachets | | | | | | | X |

**Fig. 3 | Study flowchart.** Caption: This flowchart was previously published in the study protocol[70]. Reproduced with permission from Ni Lochlainn et al.[70].

anabolic resistance. We also illustrate the feasibility of remotely delivered trials for older people, which holds promise for future studies in this area, aiming to reduce the under-representation of older people in clinical trials and reduce research costs.

## Methods

The study design is described in detail in the previously published protocol[69]. In brief, the PROMOTe (effect of PRebiotic and prOtein on Muscle in Older Twins) trial was a randomised controlled trial in which twin pairs (monozygotic and dizygotic) were randomised, one twin to each study arm. Both twins consumed a protein (BCAA) supplement powder, and in one twin from each pair, this was combined with a prebiotic supplement (inulin and fructo-oligo-saccharides) and in the other twin from each pair, it was combined with a placebo (maltodextrin). These pre-mixed supplements were in identical sachets, and each participant was advised to take one sachet a day for 12 weeks in a glass of water or another drink at the same time each day. We advised participants in both groups to undertake resistance exercises.

This trial was delivered remotely, using postal packs to distribute the intervention and collect measures and samples from participants and various forms of technology, including video teleconferencing, online questionnaires, online cognitive testing, and online food diary software. This remote study design was chosen partly due to the COVID-19 pandemic and the associated travel restrictions, as well as potential concerns among older people about attending a hospital setting. Further, many older people can find travel difficult for a variety of reasons and the National Institute of Health Research has called for

innovative study designs to improve inclusivity of under-represented groups, including older adults[66,70]. This was facilitated by the ever-increasing internet and technological literacy of older people, which was accelerated by the COVID-19 pandemic lockdowns[71,72].

The flow of the study and measures taken are shown in Fig. 3. All participants were sent a postal pack containing all necessary apparatus and instructions to take the biological samples, including paid return envelopes, a dynamometer to measure grip strength (Kuptone; Model EH101), a four-metre-long ribbon to assist in measuring gait speed and a measuring tape for height estimation. Participants had the opportunity to ask researchers for further guidance on any element of the trial during the video teleconference visit.

Sex data was based on self-reports from participants. All twins were same-sex. Gender was not collected. Height measurements were taken by the participant with a measuring tape provided in the postal pack. Participants were asked to weigh themselves if they had a weighing scale. Weighing scales could be checked using a standard household item, such as a tin of beans. Short Physical Performance Battery (SPPB) was carried out remotely (this includes chair-rise time), with real-time instructions from a trained researcher. Handgrip strength was measured using the provided dynamometer, with guidance from the trained researcher, during the video call. Stool samples were collected by the participants themselves using the sample collection kits provided. Twins were asked to collect a "pea-sized" stool sample into a DNA/RNA Shield Faecal Collection Tube (Zymo Research), and these were posted to the laboratory. Upon reception, samples were processed using glass beads, vortexing, centrifugation and aliquoting of supernatant for temporary and long-term storage at

−80 °C. Microbial DNA extraction was performed on all samples using a customised MagMax Core Nucleic Acid Purification Kit and MagMax Core Mechanical Lysis Module. The protocol was optimised for 1 g/ml starting weight faeces and final 100 μl elution volume. Subsequent DNA was stored at −20 °C for the long term. Microbial DNA was sent to *Clinical Microbiomics Ltd.* for gut microbiota characterisation using shotgun metagenomic sequencing (see Supplementary Note 1 for detailed methodology).

Cognitive tests from the CANTAB battery included tests of executive function, namely the One Touch Stockings of Cambridge and spatial working memory tests, and tests of memory, namely spatial span, pattern recognition memory, and paired associates learning[40]. Three-day food diary data were entered and analysed on the *MyFood24* online software[73,74].

Participants were considered compliant to supplement if they answered the question 'Have you been taking your supplements daily?' with either 'Yes−everyday' or 'Almost every day' and were considered compliant to exercise if they reported undertaking the exercises twice per week or more. Lastly, participants were asked to keep the remaining sachets at their final video visit for these to be counted. This number was compared to the number of days since the participant started the study, and a percentage adherence was calculated. This study was registered with ClinicalTrials.gov (NCT04309292).

## Inclusion/exclusion criteria

Participants were eligible for inclusion if they were aged 60 years or older and had previously reported low dietary protein intake (<1 g/kg body weight/day) according to the European Society for Clinical Nutrition and Metabolism (ESPEN) guidance for older adults[8]. This was chosen over the Recommended Nutrient Intake (RNI) for protein in the UK (0.8 g/kg/day) due to anabolic resistance of skeletal muscle associated with ageing. Intake of protein to determine entry to the study was measured a priori within the TwinsUK cohort. In addition, participants had to have access to a computer or tablet device, in order to be able to complete the remote visits via video teleconference. The exclusion criteria included severe food allergy, current or recent (preceding 3 months) use of antibiotics, protein supplements, prebiotics or probiotics, chronic kidney disease stage 3 or higher, weight loss of ≥5% of body weight in the preceding 12 months, any significant injury or surgery which currently affects physical functioning, and current involvement in other interventional studies. These criteria were selected to avoid contraindications to the interventions, to avoid contamination of data collected, for example, with recent use of a protein supplement, and to maximise the reliability of the gut microbiota samples collected, for example, by excluding recent antibiotic use and significant gastrointestinal disease.

## Study Interventions

Participants were provided with sachets of food supplements in powder form. All sachets contained 3.32 g of branched-chain amino acid protein powder, consisting of L-leucine 1660 mg, L-isoleucine 830 mg, and L-valine 830 mg. The intervention arm sachets also contained 7.5 g of prebiotic (Darmocare Pre®, Bonusan), which consists of inulin (min. 3375 mg) and fructo-oligosaccharides (FOS) (min. 3488 mg). The placebo sachets contained 7.5 g of maltodextrin powder. All of these food supplements are available commercially without prescription. The choice of prebiotic was based on a trial using the same intervention, which showed improved handgrip strength and exhaustion levels, two of the Fried frailty criteria in an older population[25]. Maltodextrin is commonly used as a placebo in the gut microbiome and dietary intervention studies[39].

The sachets for both arms were manufactured to be contained in indistinguishable sachets, identifiable by a random number, which was accessible only by the King's Clinical Trial Unit (KCTU), which carried out the randomisation for the study. Randomisation and arm allocation were done remotely by the KCTU, who issued the corresponding number for the correct sachets to be sent to each participant by the research team. Randomisation was done as twin pairs, each pair as a fixed block of two−one twin in each pair randomly allocated to each arm. All participants and researchers were blinded until data analysis was complete.

All participants were asked to retain any remaining sachets at the end of the trial period as a measure of compliance. Lastly, all participants were encouraged to engage in regular resistance exercise at least twice per week throughout the intervention and were provided with written advice regarding this at the beginning of the study, available from the NHS website: https://www.nhs.uk/live-well/exercise/strength-and-flexibility-exercises/strength-exercises/[75]. The strength exercises included squats, calf raises, sit-to-stand, leg lifts, leg extensions, wall press ups and bicep curls. These exercises are routinely advised for older adults, can be done in their own homes, and do not represent a substantial exercise programme. Text message reminders were sent weekly throughout the intervention period to encourage compliance with exercises and supplement sachets.

## Outcomes

Outcomes were collected at the study visits via video teleconferencing, via remote completion of questionnaires, a 3-day food diary to measure protein and other dietary intake to account for any differences in intake that would lead to confounding and cognitive test battery, and via postal receipt of biological samples from participants. The primary outcome was a change in chair rise time (time to do 5 chair rises without using arms), measured at baseline and study end. Chair rise time is a component of the short physical performance battery and is associated with quality of life, physical function, frailty, multimorbidity and, indeed, cognitive function[76,77]. It is the recommended measure of the strength in the European Working Group of Sarcopenia of Older Persons 2 guidance[67].

Secondary outcomes included cognitive battery factor score, SPPB score (includes chair rise time and gait speed), grip strength, gait speed, self-reported physical activity levels using the International Physical Activity Questionnaire (IPAQ) and Simplified Nutritional Assessment Questionnaire (SNAQ) appetite score.

## Gut microbiome

The full methodology of the gut microbiome characterisation is provided in Supplementary Note 1. In summary, a total of 137 DNA aliquots were sequenced to an average depth of 21.6 million (M) read pairs (Illumina 2 × 150 PE) per sample. Host contamination was discarded, adapters and bases were removed by trimming, and read pairs in which post reads passed filtering with a length of at least 100 bp were retained; these were classified as high-quality non-host (HQNH) reads. 82.8% of the HQNH reads were mapped to the Clinical Microbiomics human gut gene catalogue using BWA mem (v. 0.7.17)[78]. 326 Metagenomic Species (MGSs) were detected on average per sample. To taxonomically annotate an MGS, its genes were blasted against NCBI RefSeq prokaryotic genomes (2022-01-19), and nt (2021-08-03) databases and rank-specific annotation criteria were used. Rarefied MGS abundance profiles were calculated by random sampling, without replacement, of a fixed number of signature gene counts per sample and then followed the procedure described above. In this study, 431127 signature gene counts were sampled. Functional annotation and profiling of the microbiome were undertaken using EggNOG mapper (v. 2.0.1)[79], Diamond mode, and KEGG modules (v. 78.2)[80].

All microbiome analyses used relative abundance for group comparisons, while alpha and beta diversity estimates were calculated from rarefied abundance matrices created by random sampling of reads without replacement. Alpha diversity refers to within-sample

diversity, while beta diversity measures are estimates of the similarity/dissimilarity between samples[81]. Within each data type (e.g., gene, MGS), all samples were represented by the same number of informative sequencing reads: rarefaction of MGS abundance was performed by sampling only from reads mapping to MGS' signature genes, and rarefaction of Kyoto Encyclopaedia of Genes and Genomes (KEGG) orthology (KO) abundance was performed by sampling only from reads mapped to a gene with an assigned KO. However, rarefaction of gene abundance was performed by sampling reads mapped to the entire gene catalogue. Alpha diversity was calculated as the number of microorganisms detected (richness), as the Shannon index based on natural logarithm, or as Faith's phylogenetic diversity. Beta diversity was calculated as the Bray–Curtis dissimilarity and weighted UniFrac distances.

The similarity between twin pairs was investigated using the Mann–Whitney $U$ test and comparing it to non-twin pairs since multiple endogenous and environmental factors, such as diet, geography, type of living, cohabitation status, and host genetics influence host microbiome[82]. Formal heritability analysis was carried out using the ACE model as described in[52], to address if the microbiota similarity between twins could be attributed to the heritability of microbial taxa.

## Sample size calculation

From existing data within the TwinsUK cohort, we observed that chair-rise time was approximately log-normal, with log10(chair rise time) having a SD of 0.126. We considered a relative reduction in chair rise time of 20% to be both plausible and clinically important. We based this on previous studies using chair rise time[35–37], however, it was noted that no study had investigated this in the context of a gut microbiome-focused intervention. Based on these figures, we computed that we would need complete data on 28 participants per group (56 in total) for 80% power. Allowing for 20% dropouts, we needed $n = 70$ participants recruited (35 per group). Using twins increases study power due to close matching at baseline, with reduced genetic and/or environmental variability, and therefore reduced confounding[83].

## Statistical analysis

Analysis was performed using Stata SE version 15.1[84], other than gut microbiome analysis, which was performed using R version 4.2.1[85]. Unmasking of the randomisation groups was only carried out after all statistical analysis was complete. A two-sided $P$ value of <0.05 was considered significant for all analyses. Participants were not excluded if they were missing baseline covariate data to satisfy the principles of an intention-to-treat (ITT) approach. Thus, for missing data in this category, mean imputation techniques were used. Alternatively, for some variables, the most recent value from the longitudinal TwinsUK cohort data was imputed if these were deemed to be within an appropriate period. Decisions on imputation methods were made prior to unblinding.

In terms of missing outcome data, for the purpose of the main analysis, it was assumed that missing data was missing at random, and the effect of the intervention was the same in those with and without the observations. No evidence was found of an imbalance of missing data within each treatment allocation.

To compare those who declined or were ineligible to take part, to those who took part, existing TwinsUK longitudinal data were used. Two-sided paired t-tests were used to compare each group. The difference between the two groups in terms of sex was adjusted for age, and likewise, the difference between the two groups in terms of age was adjusted for sex to assess whether one of these variables was driving variation between the two groups.

To characterise the differences between study arms at baseline, continuous variables were compared using two sample t-tests and categorical variables with Pearson's chi-squared tests. Two-sided paired t-tests were used to compare dietary data between baseline and study end.

To investigate cognition overall, a factor analysis score was used to combine the results of the five cognitive tests (Supplementary Note 2). Factor loadings from the baseline were applied to both the baseline and study-end cognitive test results to ensure uniform loadings. One factor was identified from the scree plot as accounting for maximum variance. Thus, a factor score at baseline and study endpoint was derived and subsequently used in the mixed effects models. Linear mixed-effects models were used to compare intervention groups (arm 1 vs arm 2; blinded) on their change in each outcome by using the measure of that outcome at the study end and including the baseline measure in the model. Twin clustering was considered as random effect, both family identifier and zygosity, and treatment group a fixed effect. Two-sided paired t-tests were used for within-group comparisons between baseline and end of intervention.

To check nutrient intake for over- and under-reporting, Goldberg cut-offs for energy intake compared to calculated basal metabolic rate ratio were used[86,87] and multiplied by a factor of 1.4 to account for physical activity[88]. The occupational activity level 'light' was chosen based on the IPAQ scores of participants at baseline.

Gut microbiota features (composition, function, diversity metrics, GMMs, and GBMs) were compared between prebiotic and placebo study arms at baseline and study end using the Wilcoxon signed-rank test. In addition, intra-group comparisons between baseline and endpoint were carried out for each arm separately. We tested whether any microbiota features were associated with the primary outcome chair-rise time either at baseline, study end or the change between the two. The change in microbiome feature was inputted into a correlation analysis versus the change in chair rise time and the change in cognitive factor score. All change variables are the difference between the baseline and study endpoint. To compute the changes in microbiota features, raw relative abundances were used, and the resulting delta values were then centred log ratio transformed and used as input for the linear models.

Pairwise comparisons of microbiota (dis)similarity were compared using the Mann-Whitney U test. Permutational multivariate analysis of variance (PERMANOVA) tests were performed using the adonis2 function from the vegan R package[89] with 1000 permutations. Analyses were adjusted for multiple tests using the Benjamini–Hochberg method to control the false discovery rate at a level of 10%. Broad-sense heritability was determined with the ACE model (from R package mets[90,91]). The model estimates how much phenotypic variation stems from additive genetic effects (A), common environment (C) and unique environment for each twin (E). The model was fitted separately to the abundances of taxa found in more than 2 samples in each arm ($abundance \sim 1$) and used to extract broad-sense heritability. More detail on the gut microbiome statistical analysis is available in Supplementary Note 1.

## Reporting summary

Further information on research design is available in the Nature Portfolio Reporting Summary linked to this article.

## Data availability

The gut microbiome sequencing data generated in this study have been deposited in the European Nucleotide Archive (ENA) at EMBL-EBI under accession number PRJEB72531. All other data are available on request from TwinsUK, and access can be obtained after approval by the TwinsUK Resource Executive Committee by following this link: https://twinsuk.ac.uk/resources-for-researchers/access-our-data/.

## Code availability

Stata code was utilised for data cleaning and analysis using Stata SE version 15.1[84]. Gut microbiome analysis utilised R version 4.2.1[85], using

R packages vegan (available from: https://cran.r-project.org/web/packages/vegan/index.html) and Mets (available from https://cran.r-project.org/web/packages/mets/index.html).

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

## Acknowledgements

The authors would like to acknowledge the National Institute of Health Research for funding this research. We would also like to acknowledge the research staff at the Department of Twin Research and all the participants. We particularly would like to thank Deborah Hart, Ayrun Nessa, Alyce Sheedy, and Gulsah Akdag who contributed a huge amount to the running of the PROMOTe trial. We thank the TwinsUK Volunteer Advisory Panel for their insightful comments and feedback regarding the study design. We would also like to thank Roosa Varjus for contributing to the microbiome analysis. This study was co-sponsored by Kings College London and Guy's and St Thomas' NHS Trust. Neither sponsor had any role in study design, data collection and analysis or manuscript writing. Funding MNL is supported by an NIHR Doctoral Fellowship (grant code: NIHR300159). C.J.S. receives funds from the Medical Research Council (MRC), Wellcome Trust (grant code WT 212904/Z/18/Z), and the Chronic Disease Research Foundation. K.W. has received funds from the MRC, NIHR, Crohn's and Colitis UK, Kenneth Rainin Foundation, Leona M. and Harry B. Helmsley Charitable Trust, Almond Board of California, Danone, and International Dried Fruit Council. TwinsUK is funded by the Wellcome Trust, Wellcome LEAP, the Medical Research Council, Versus Arthritis, Engineering & Physical Sciences Research Council, Biotechnology and Biological Sciences Research Council, European Commission, Chronic Disease Research Foundation (CDRF), Zoe Ltd., the National Institute for Health and Care Research (NIHR) Clinical Research Network (CRN) and Biomedical Research Centre based at Guy's and St Thomas' NHS Foundation Trust in partnership with King's College London. The myfood24 dietary reporting software was used in this study. myfood24 was developed through Medical Research Council funding, grant G110235. myfood24 is now being supported by spinout company Dietary Assessment Ltd. Requests to use myfood24 should be made to enquiries@myfood24.org.

## Author contributions

M.N.L., K.W. and C.J.S. conceived the study and wrote the protocol with input from R.C.E.B., M.P.G., S.W., A.F.B, A.N., A.S., G.A., D.H., C.M., S.D.R.H., A.A.W. and C.G., A.N., A.S., G.A., D.H. and C.M. supported the clinical trial, including recruitment and/or management of patients in the trial. M.N.L. performed the statistical analysis with input from C.J.S, K.W. R.C.E.B, P.T.S. and G.R. J.M.M. carried out the microbiome analysis with input from M.N.L., C.J.S. and R.C.E.B. M.N.L., S.W., A.F.B, A.N. and A.S. helped collect data and samples. M.N.L. wrote the paper. All the authors reviewed the paper and approved the final version.

## Competing interests

C.J.S. has consulted for Zoe Limited for work on the Zoe Health Study. Janne Marie Moll is an employee of Clinical Microbiomics. K.W. is a co-inventor/patent holder for volatile organic compounds in the diagnosis and management of irritable bowel syndrome. The remaining authors declare no competing interests.

## Ethical approval

The PROMOTe Study was approved by NHS North of Scotland Research Ethics Service (REC reference 21/NS/0045), IRAS ID 257415. This study was carried out under TwinsUK BioBank ethics, approved by North West —Liverpool Central Research Ethics Committee (REC reference 19/NW/0187), IRAS ID 258513. This approval supersedes earlier approvals granted to TwinsUK by the St Thomas' Hospital Research Ethics Committee, later London—Westminster Research Ethics Committee (REC reference EC04/015), which have now been subsumed within the TwinsUK BioBank. Written informed consent was obtained from all participants. All research, therefore carried out in accordance with the ethical standards laid down in the 1964 Declaration of Helsinki and its later amendments.

## Additional information

¹King's College London, Department of Twin Research and Genetic Epidemiology, London SE1 7EH, UK. ²The Alan Turing Institute, London NW1 2DB, UK. ³Clinical Microbiomics, Copenhagen, Denmark. ⁴GKT School of Medical Education, King's College London, London, UK. ⁵Unit for Medical Statistics/ Department for Women and Children's Health, School of Life Course and Population Sciences, Faculty of Life Sciences and Medicine, King's College London, London, UK. ⁶King's Clinical Trials Unit, Research Management and Innovation Directorate, King's College London, London, UK. ⁷Centre for Human & Applied

Physiological Sciences, King's College London, London, UK. [8]Department of Epidemiology and Public Health, Norwich Medical School, University of East Anglia, Norwich NR4 7TJ, UK. [9]School of Sport, Exercise, and Rehabilitation Sciences, University of Birmingham, Birmingham, UK. [10]MRC-Versus Arthritis Centre for Musculoskeletal Ageing Research, Birmingham, UK. [11]NIHR Birmingham Biomedical Research Centre, University Hospitals Birmingham NHS Foundation Trust and University of Birmingham, Birmingham, UK. [12]King's College London, Department of Nutritional Sciences, Franklin Wilkins Building, SE1 9NH London, UK. ✉e-mail: mary.ni_lochlainn@kcl.ac.uk; claire.j.steves@kcl.ac.uk

