## [Peer Review File · Nature Communications]

Effect of gut microbiome modulation on muscle function and cognition: the PROMOTe randomised controlled trialREVIEWER COMMENTS

Reviewer #1 (Remarks to the Author):

The manuscript describes a clinical trial investigating the possible added health benefits of inulin supplementation to BCAA+Exercise intervention in older adults through changes in gut microbiota composition. The major strengths of the study are the sample size of 36 twin pairs and adaptation of the methodology to facilitate remote participation. Such approaches are essential to promote inclusivity of older adults into biomedical research to facilitate the design of new treatments for common age-related health conditions. However, there are specific weaknesses in the study design and the simple correlation analysis fails to provide novel mechanistic insight into how changes in gut microbiota composition would drive improvements in muscle/cognitive health.

Specific comments

- 1) The authors state in the introduction that “many older adults are unable and/or unwilling to undertake a substantial exercise regimen. Thus, other intervention options are needed”. I was therefore surprised that the clinical trial then involved all older participants being asked to engage in regular resistance exercise throughout the 12-week intervention. The description of the prescribed resistance exercise is limited. Were all participants naïve to this type of resistance exercise at baseline or were some habitual exercisers? Such variability would have had substantial impact on primary/secondary outcomes. The description of compliance measurement to the resistance exercise is also limited. Self-reported compliance to prescribed/free-living exercise training is notoriously unreliable, and outcomes have greater inter-individual variability compared to supervised/lab-based exercise training.
- 2) The presented rationale for investigating links between the gut microbiota and anabolic resistance is weak. The authors use their own review article (Ref 6) to briefly list possible mechanistic links between the gut microbiota and anabolic resistance, but there is no actual primary evidence that individual differences in the gut microbiota have a causative impact on rates of muscle protein synthesis in response to an anabolic stimulus (protein intake or exercise). The present manuscript doesn't add to this important missing preliminary step.
- 3) The rationale for using inulin as a ‘prebiotic’ is never explained. The study aims to modulate the gut microbiome, however the impact of inulin in humans is rather limited. Indeed, a recent meta-analysis indicates that inulin supplementation would only reliably alter abundance of ~5 taxa (<https://pubmed.ncbi.nlm.nih.gov/31707507/>). It is never explained why targeting these specific taxa with the chosen prebiotic intervention would improve primary/secondary outcomes. Are these taxa reduced in older adults with sarcopenia/cognitive impairment?
- 4) Whilst inulin can be ‘jazzed up’ and described as a prebiotic, it is a common dietary fibre supplement. The manuscript provides information on dietary changes at baseline and follow-up in the entire cohort and separate genders (Table S1-2) but does not present changes for the placebo and prebiotic groups. How did the interventions alter total macro- and micronutrient intake (including fibre) within each

group. Importantly, total energy intake is reduced in response to intervention in the entire cohort (S1-2), but the possible impact this had on outcome measures and whether this was driven by changes in one intervention group is not discussed.

5) The authors do acknowledge that the power calculation for the primary outcome (chair-rise time) may have been inadequate. However, I am unsure how they ever thought a 20% improvement in response to a 12-week intervention period would ever be plausible. The references used to support this power calculation appear inadequate for this clinical trial. Two references are cross-sectional/prospective studies associating chair-rise time with health outcomes (Ref 28 and Ref 30), whilst the only clinical trial involves a physical activity intervention of far longer duration (12 months).

6) The results section states the prebiotic intervention impacts both gut microbiota composition and function. However, the functional assessments from metagenomic analysis are limited to the 'functional potential' of the gut microbiota. Complimentary metabolite analysis of faecal water would have confirmed that changes in the pentose phosphate pathway does alter the generation/bioavailability of precursors for amino acid biosynthesis. This information would also have advanced mechanistic insight.

7) Inulin has an EFSA health claim to promote beneficial effects on bowel function. Were changes in bowel frequency measured in the cohort of older adults?

8) The choice of maltodextrin as the 'placebo' for inulin supplementation is not explained or justified. Maltodextrin is not an inert supplement; the addition of 7.5 g of rapidly digestible glucose polymer in the placebo group would itself generate physiological signals (e.g. higher insulin) that could influence outcome measures. For example, the combination of amino acids with carbohydrate (i.e. the placebo) has been shown to promote greater increases in muscle protein synthesis than amino acids alone (i.e. the prebiotic) (<https://pubmed.ncbi.nlm.nih.gov/12618575/>). The choice of 'placebo' supplements is therefore fundamentally flawed when targeting muscle function as a primary outcome measure. Furthermore, a higher glycaemic load is associated with poor cognitive performance in older adults (<https://pubmed.ncbi.nlm.nih.gov/25034880/>).

Reviewer #2 (Remarks to the Author):

In this placebo controlled double blinded randomized controlled trial of 36 twin pairs (72 individuals), aged 60 and over, each twin pair was block randomized to receive either branched chain amino acid (BCAA) supplementation plus placebo or BCAA supplementation plus a prebiotic (7.5 gram of inulin/FOS) gut microbiome modulator daily for 12 weeks. Regular resistance exercise was prescribed to all participants. The outcomes were physical function and cognition. The trial was carried out remotely using video visits, online questionnaires, food diaries and cognitive testing, and posting of equipment and biological samples. Authors reported that the prebiotic supplement was well tolerated (although 22% reported side effects) and did result in a changed gut microbiome [e.g., increased Bifidobacterium].

There was no significant difference between prebiotic and placebo for the primary outcome of chair rise time ($\beta=0.579$; 95% CI -1.080-2.239 $p=0.494$). The Prebiotic improved cognition (first factor score versus placebo ($\beta=0.482$; 95% CI 0.141-0.823; $p=0.014$)). Authors concluded that the results demonstrate that cheap and readily available gut microbiome interventions may improve cognition in ageing population. They also concluded that the results showed the feasibility of remotely delivered trials for older people.

This is a carefully designed study with multiple strengths including strong scientific premise, use of twins, well-defined outcome measures, use of appropriate data analysis methods, well-reasoned selected prebiotic dose, use of BCAA and study duration of 12 weeks. Conclusion is supported by the reported results.

I have the following questions/comments:

1- It is stated that prebiotic increased Bifidobacterium taxa. It should of course underline that increased (or decreased) in abundance of any bacteria taxa like increased Bifidobacterium is relative increased abundance

2- The current definition of prebiotic is no longer defined by increase in Bifidobacterium or lactobacillus – it is now defined by increased abundance of “anti-inflammatory” /protective bacteria taxa. Here the authors imply that inulin/FOS had prebiotic effects since Bifidobacterium was increased. This point should be stated in the discussion where the authors discussed lack of impact of prebiotic on muscle strength. Indeed, one potential explanation is that inulin/FOS might not be the appropriate prebiotic (microbiota modulator) for muscle strength in elderly. Prior studies reported that inulin/FOS might have a pro-inflammatory effect which could mitigate their microbiota modifying effects for change in muscle strength.

3- The correlation between Bifidobacterium and improved cognition in prebiotic arm is confusing. It is reported there is a negative correlation between Bifidobacterium and improved cognition and yet prebiotic increased Bifidobacterium and improved one marker of cognition. This requires explanation. Also, it appears that there is no correlation between Bifidobacterium and cognition in placebo arm . was there any correlation at baseline? If indeed Bifidobacterium plays a role in cognition, there should also be a correlation at baseline too.

4- It is rather surprising that the microbiota community at the end of the study was not dissimilar in twin pair when one had prebiotic and another had placebo, considering that as a group prebiotic modified microbiota community. I acknowledge, as authors stated, that the core microbiota community is established in the first 3 years of life (inheritance, vertical mother/infant transfer of microbiota and early life events) , but still environmental events in later life (e.g. antibiotics, infection, colonoscopy, stress, diet) profoundly (at least temporarily) impact microbiota community even in twins. Could lack of dissimilarity at the end of 12 weeks suggest partial recovery of prebiotic-induced microbiota community changes? This requires additional discussion.

Reviewer #3 (Remarks to the Author):

In light of the global demographic shift towards an aging population, a deeper understanding of cognitive shifts in the elderly has become paramount for researchers in this domain. Lochlainn et al. present a rigorous randomized controlled trial (RCT) evaluating the potential of gut microbiome modulation via prebiotics to enhance muscle functionality. The clinical implications and relevance of this study are articulated with precision. The utilization of twin pairs in the study design strengthens the internal validity of the trial. Notably, the remote delivery of this trial underscores a significant innovation, facilitating the inclusion of often under-represented segments of the elderly population. Nevertheless, I harbor major reservations concerning the statistical methodologies employed and the ensuing results.

Major comments

1. The manuscript contains a discrepancy regarding the statistical tests applied to beta diversities. Specifically, lines 744 and 795 reference the Mann-Whitney U test, yet lines 796-797 make mention of the PERMANOVA test. I would kindly request clarification on which statistical method was employed in the context of Figure E4 and E5.

2. Although the primary focus of this study is not solely on the microbiome analysis between treatment groups, and it largely serves an exploratory analysis, it's crucial to emphasize the need for a more rigorous analysis plan, especially given the innovative nature of this study.

(1) The manuscript lacks crucial details regarding the form of microbial abundance used. Was relative abundance (or proportions) employed, or were raw counts analyzed? It's a well-established fact that microbiome data are inherently compositional. Directly comparing raw counts using standard statistical tests (like the t-test) would be statistically invalid. Conversely, comparisons involving relative abundance can be nuanced and pose interpretative challenges. Any change in the absolute abundance of one taxon has ramifications for the overall microbial profile. Consequently, relative abundance is often used for a global test. It would be highly advisable for the authors to review all microbiome features incorporated in this study and refine their analytical approaches accordingly.

(2) There appears to be an absence of a comprehensive differential abundance (DA) analysis, even though Figure 2 seems to touch upon this. In the context of a mixed-effects model, I'd suggest considering more sophisticated tools like LinDA (<https://cran.r-project.org/web/packages/MicrobiomeStat/index.html>), ANCOM-BC2 (<https://bioconductor.org/packages/release/bioc/vignettes/ANCOMBC/inst/doc/ANCOMBC2.html>), MaAsLin2 (<https://huttenhower.sph.harvard.edu/maaslin/>), or other available methods.

3. Sample size calculation needs clarifications. When I attempted to verify the sample size through manual computation, considering an SD (σ) of 0.126 and mean difference (Δ) equal to $\log_{10} 0.8$, my results were different from what's presented in the manuscript.

$$n = \frac{2\sigma^2(Z_{1-\alpha} + Z_{1-\beta})^2}{\Delta^2} = 21 \text{ per group, one-sided}$$

$$n = \frac{2\sigma^2(Z_{1-\alpha/2} + Z_{1-\beta})^2}{\Delta^2} = 27 \text{ per group, if two-sided}$$

Additionally, when I utilized the `pwr.t.test` function in R, the outputs were again slightly varying:

```
> pwr.t.test(d = log(0.8, 10)/0.126, sig.level = 0.05, power = 0.8, type = "two.sample", alternative = "two.sided")
```

```
Two-sample t test power calculation
```

```
      n = 27.53045
      d = 0.7691271
sig.level = 0.05
  power = 0.8
alternative = two.sided
```

NOTE: n is number in *each* group

```
> pwr.t.test(d = log(0.8, 10)/0.126, sig.level = 0.05, power = 0.8, type = "two.sample", alternative = "less")
```

```
Two-sample t test power calculation
```

```
      n = 21.61221
      d = -0.7691271
sig.level = 0.05
  power = 0.8
alternative = less
```

NOTE: n is number in *each* group

On another note, assuming the mentioned sample size of 30 per group is accurate, by factoring in the anticipated 20% dropouts, it would necessitate recruiting 37.5 individuals per group, not 35. I might be missing some context or specific details, so I'd appreciate any clarifications on this matter.

4. Reproducibility. I noticed that while the software utilized was provided, the specific code or scripts employed for the analyses were not shared. Making this available would greatly enhance the transparency and allow for more in-depth assessment and reproduction of the findings.

Minor comments

1. There seems to be a discrepancy between Table 3 and line 137 in the main text. The coefficients and 95% CI for the Cognition factor score in Table 3 are presented as -0.48 and [-0.81, 0.141], respectively. However, line 137 in the main text depicts all values as positive. Kindly verify and rectify for consistency.

2. I observed that lines 137-138 in the main text do not include the coefficient, 95% CI, and p-value for PAL. To ensure comprehensive representation, it would be helpful to have these details included. Moreover, please ensure that the signs used for these values are consistent with those in other sections of the paper.

3. To maintain terminological consistency, I suggest changing the title of Figure E5 from "Wilcoxon test" to "Mann-Whitney U test."

4. Regarding Figure 2, it would be beneficial to differentiate the two groups more clearly by opting for more contrasting color sets, perhaps red and blue. Additionally, the directionality of the "Change between baseline & study end" on the x-axis could be made more explicit. If I understand correctly from line 182, it should represent the difference calculated as "study end minus baseline".

Effect of gut microbiome modulation on muscle function and cognition: the PROMOTe randomised controlled trial: Responses to Reviewers

Reviewer 1

Reviewer Comment	Response
The manuscript describes a clinical trial investigating the possible added health benefits of inulin supplementation to BCAA+Exercise intervention in older adults through changes in gut microbiota composition. The major strengths of the study are the sample size of 36 twin pairs and adaptation of the methodology to facilitate remote participation. Such approaches are essential to promote inclusivity of older adults into biomedical research to facilitate the design of new treatments for common age-related health conditions. However, there are specific weaknesses in the study design and the simple correlation analysis fails to provide novel mechanistic insight into how changes in gut microbiota composition would drive improvements in muscle/cognitive health.	Many thanks for these positive comments. We very much appreciate you taking the time to review our manuscript.
Specific comments	
1) The authors state in the introduction that “many older adults are unable and/or unwilling to undertake a substantial exercise regimen. Thus, other intervention options are needed”. I was therefore surprised that the clinical trial then involved all older participants being asked to engage in regular resistance exercise throughout the 12-week intervention. The description of the prescribed resistance exercise is limited. Were all participants naïve to this type of resistance exercise at baseline or were some habitual exercisers? Such variability would have had substantial impact on primary/secondary outcomes. The description of compliance measurement to the resistance exercise is also limited. Self-reported compliance to prescribed/free-living exercise training is notoriously unreliable, and outcomes have greater inter-individual variability compared to supervised/lab-based exercise training.	Many thanks for your comments. We have now improved the wording of this section of the introduction. As described in Section 11.2 Study Interventions, the prescribed exercises are from the website of the National Health Service (NHS) in the UK. https://www.nhs.uk/live-well/exercise/strength-and-flexibility-exercises/strength-exercises/ These are routinely advised exercises for older adults and do not represent a substantial exercise regimen. We have now amended this section of the methods to make this clearer. While individual exercise detail was not captured, physical activity information was collected via the International Physical Activity Questionnaire (IPAQ) – there were no significant differences between the two study arms at baseline in terms of IPAQ. Further,

	there was no significant difference between the two arms for IPAQ activity levels in the linear mixed effects models analysing change in activity level across the study intervention period, as shown in Table 3. We agree that lab-based or supervised exercise can be monitored more closely, however this was not feasible in this remotely delivered trial. In our trial, we encouraged compliance with exercise advice using motivational text messages sent to participants during the intervention period. Further, we aimed to carry out a translational trial, with an intervention which could be feasible in an older population at scale, without huge associated costs. Travel to a lab or gym for supervised exercise is not always feasible for older people, and resources for staffing such programmes can be limited. Lastly, the covid pandemic was a major factor in our minds when developing the study design and protocol, and supervised exercises at the time were not feasible with so many unknowns ahead. Thus, with all the above in mind, we chose home-based self-directed exercises.
2) The presented rationale for investigating links between the gut microbiota and anabolic resistance is weak. The authors use their own review article (Ref 6) to briefly list possible mechanistic links between the gut microbiota and anabolic resistance, but there is no actual primary evidence that individual differences in the gut microbiota have a causative impact on rates of muscle protein synthesis in response to an anabolic stimulus (protein intake or exercise). The present manuscript doesn't add to this important missing preliminary step.	Many thanks for your comments. We have added a number of other references to the introduction, which all discuss links between the gut microbiota and anabolic resistance. It was outside the scope of our clinical trial to study the mechanistic links between the gut microbiota and anabolic resistance. Rather we aimed to test whether utilising a prebiotic food supplement to influence the gut microbiota could lead to a change in muscle strength, suggesting a potential influence on anabolic resistance by proxy. Indeed, our aim was to carry out a translational trial, with outcomes directly related to potential patient benefit. Unfortunately, we did not have the available resources to do additional mechanistic work, though we can see that such research would be of interest.

3) The rationale for using inulin as a 'prebiotic' is never explained. The study aims to modulate the gut microbiome, however the impact of inulin in humans is rather limited. Indeed, a recent meta-analysis indicates that inulin supplementation would only reliably alter abundance of ~5 taxa (https://pubmed.ncbi.nlm.nih.gov/31707507/). It is never explained why targeting these specific taxa with the chosen prebiotic intervention would improve primary/secondary outcomes. Are these taxa reduced in older adults with sarcopenia/cognitive impairment?	Thank you for your comments. The prebiotic utilised in this trial was a combination of inulin and fructo-oligosaccharides (FOS) (another inulin-type fructan) as described in Section 11.2 'Study Interventions'. Inulin-type fructans were the first prebiotics ever to be discovered, are the most widely used prebiotics in clinical trials, and most widely used prebiotic food supplement. A key feature of prebiotic supplementation is that effects on global composition of microbiome are rarely seen, instead, as you say these are isolated to a small number of taxa (e.g. bifidobacteria). As well as modifying these taxa, prebiotic supplementation also can modify microbiome metabolism, including the production of SCFA. Importantly, SCFA production by the gut microbiome has been associated with anabolism. Further detail on rationale for the choice of this specific prebiotic combination has been added to Section 11.2 of the manuscript to ensure this is clear. Alternative approaches to microbiome modification have other challenges. First, probiotics also do not impact global microbiome composition, instead microbiome changes are limited to solely those contained within the supplement itself. Second, although faecal microbiome transplant makes profound changes to recipient microbiome, this is a highly invasive intervention that may not have been suitable for many participants.
4) Whilst inulin can be 'jazzed up' and described as a prebiotic, it is a common dietary fibre supplement. The manuscript provides information on dietary changes at baseline and follow-up in the entire cohort and separate genders (Table S1-2) but does not present changes for the placebo and prebiotic groups. How did the interventions alter total macro- and micronutrient intake (including fibre) within each group. Importantly, total energy intake is reduced in response to intervention in the entire cohort (S1-2), but the possible impact this had on	Many thanks for your comments. We have now included an additional Table (Table S3) which presents changes in energy and nutrient intake for the placebo and prebiotic groups. There were no differences between prebiotic and placebo groups at baseline or at study end point for starch, non-starch polysaccharides (NSP), or total fibre intake (AOAC). There was also no within group differences between baseline and study end for either

outcome measures and whether this was driven by changes in one intervention group is not discussed.	arm, for oligosaccharides, starch, non-starch polysaccharides, or fibre (AOAC). The total energy intake reduction overall did appear to be driven by the prebiotic intervention arm. The average reduction in energy intake between baseline and study end was 132.4 kcal/d in the prebiotic group, which as you suggest may be a prebiotic-effect or fibre-effect on appetite regulation. In the placebo group the average reduction in energy was only 56.8 kcal/d. That leaves an average difference of 75.6 kcal/d between the two arms. The standard deviations were 339.6-519.5 kcal/d for these comparisons (Table S3). Thus, while there is statistical significance when comparing energy intake between baseline and study end for the prebiotic group, we suggest that 75.6 kcal/d is unlikely to be of any biological significance. Further we suspect this value would fall within measurement error limits for energy reporting. In addition, there are no differences between baseline and study end for protein intake, in the group overall, or in either arm individually. We have included this energy result in the Results section of the paper.
5) The authors do acknowledge that the power calculation for the primary outcome (chair-rise time) may have been inadequate. However, I am unsure how they ever thought a 20% improvement in response to a 12-week intervention period would ever be plausible. The references used to support this power calculation appear inadequate for this clinical trial. Two references are cross-sectional/prospective studies associating chair-rise time with health outcomes (Ref 28 and Ref 30), whilst the only clinical trial involves a physical activity intervention of far longer duration (12 months).	Thank you for your comments. Indeed, we acknowledge in the paper that the power calculation may have been inadequate and unfortunately there were no trials of gut microbiome interventions with an outcome of muscle strength as measured by the chair stand test at the time. We suspect that a 20% improvement would represent a clinically important improvement, and indeed others have reported an improvement of 20% (e.g. DOI: 10.1016/j.exger.2014.01.027), showing such a degree of improvement in this outcome is possible, albeit in a study of a non-microbiome related intervention. A longer intervention period was not feasible for this trial, and we were also guided by the

	Buigues et al. trial which used the same prebiotic intervention and reported improved grip strength (notably a 20% improvement) and frailty index over a similar intervention period to ours. DOI: 10.3390/ijms17060932 We hope to carry out a future trial with a longer intervention period.
6) The results section states the prebiotic intervention impacts both gut microbiota composition and function. However, the functional assessments from metagenomic analysis are limited to the ‘functional potential’ of the gut microbiota. Complimentary metabolite analysis of faecal water would have confirmed that changes in the pentose phosphate pathway does alter the generation/bioavailability of precursors for amino acid biosynthesis. This information would also have advanced mechanistic insight.	Thank you for your comments. We agree this would have been interesting. We did not have funding from the NIHR to examine faecal water in this trial. Future studies could examine meta-transcriptomics or metabolomics of the stool. We have added the following additional comment on functional assessments to the study limitations: “..functional analysis of the gut microbiome in this study is based on metagenomic data rather than direct measurement. Future research could examine meta transcriptomics or metabolomics of stool to examine this further.”
7) Inulin has an EFSA health claim to promote beneficial effects on bowel function. Were changes in bowel frequency measured in the cohort of older adults?	Many thanks for your comment. We have now included an additional table in the supplementary materials (Table S4), which includes the information on participants’ bowel habits before and after intervention in both arms. There was no difference between the two arms in terms of any of the questions relating to bowel habits. We have also included this information in the Results section of the paper.
8) The choice of maltodextrin as the ‘placebo’ for inulin supplementation is not explained or justified. Maltodextrin is not an inert supplement; the addition of 7.5 g of rapidly digestible glucose polymer in the placebo group would itself generate physiological signals (e.g. higher insulin) that could influence outcome measures. For example, the combination of amino acids with carbohydrate (i.e. the placebo) has been shown to promote greater increases in muscle protein synthesis than amino acids alone (i.e. the prebiotic)	Thank you for your comment. We have updated the methods section. The study by Buigues et al. (Described in: PMID: 27314331 and PMID: 30734832) was an important influence on our study design. Their study showed that a prebiotic formulation of fructooligosaccharides and inulin improved frailty index, hand grip strength, and exhaustion levels in a cohort of older adults. They also used maltodextrin as the placebo in their trial. We aimed to replicate their findings

(<https://pubmed.ncbi.nlm.nih.gov/12618575/>). The choice of 'placebo' supplements is therefore fundamentally flawed when targeting muscle function as a primary outcome measure. Furthermore, a higher glycaemic load is associated with poor cognitive performance in older adults (<https://pubmed.ncbi.nlm.nih.gov/25034880/>).

in our design, choosing the same prebiotic formulation, and the same placebo.

Many other studies of gut microbiome interventions have used maltodextrin as a placebo. Examples include:
<https://doi.org/10.1038/s41598-019-41837-3>
<https://doi.org/10.1186/s12916-022-02299-z>
[10.3390/ijms17060932](https://doi.org/10.1038/s41598-022-02299-z)
<https://doi.org/10.1016/j.ajcnut.2023.08.016>
<https://doi.org/10.1053/j.gastro.2017.05.003>

In this meta-analysis of inulin trials in type two diabetes and obesity, maltodextrin is used as the placebo in 50% of the studies cited. (PMID: 31534973).

Indeed in our own meta-analysis of fibre interventions on gut microbiota composition: <https://doi.org/10.1093/ajcn/nqy041>, it is used as the placebo in 20 of 43 placebo-controlled trials described, while 9 others used another sugar polymer. Of the remaining trials described, it was common for researchers to use a non-fortified version of the fortified intervention, many of which contained sugar.

In another meta-analysis of inulin/FOS intervention studies, 33 trials were included. Thirteen used regular control diet as comparator. Of the remaining 20, 18 (90%) used a sugar placebo, 9 (45%) of which specifically used maltodextrin. PMID: 34401107

Many thanks for sharing the Miller et al. paper, which is a very interesting study. In their work, 35g of carbohydrate was used to show physiological signals. Unfortunately, the study does not specify exactly what carbohydrate is in the drink administered. It is notable that the carbohydrate drink given did indeed contain a range of amino acids including leucine and isoleucine. Maltodextrin does not contain any individual amino acids itself (although of course we did also provide a protein supplement to our participants).

In contrast, our study provided only 7.5 g/d of maltodextrin, the equivalent of one and a half

teaspoons, while their dose was 35 g/d (almost a 5-fold higher dose). Thus, it is not a comparable intervention to the one used in their study.

Further, many thanks for sharing the study from the group in University College Cork on glycaemic index. This interesting paper is based on routine dietary data. In terms of routine dietary data in the PROMOTe study, there was no difference between baseline and study end, overall or in either arm, for carbohydrate (baseline mean 166.7 g/d (SD :59.6 g/d), study end mean 158.4g/d (SD: 58.1g/d), p=0.115) or sugar intake (baseline mean 74.0g/d (SD: 37.5g/d), study end 69.7g/d (SD: 34.9g/d), p= 0.106), as shown in Table S1 and S3.

Maltodextrin is a glucose polymer and therefore may have increased the glycaemic load in the placebo group. However, administration of maltodextrin did not lead to a decline in cognition in the placebo group. There was a minor improvement in cognition in the placebo group (baseline cognitive factor score mean -0.23 (SD: 0.94), study end mean 0.18 (SD: 0.6)). In fact, there was an improvement in both groups, with a greater improvement in the prebiotic group (baseline cognitive factor score mean 0.05 (SD:1.00), study end mean 0.59 (SD: 0.68). P-value for linear mixed models comparing both groups = 0.014.

Reviewer 2

Reviewer Comment	Response
In this placebo controlled double blinded randomized controlled trial of 36 twin pairs (72 individuals), aged 60 and over, each twin pair was block randomized to receive either branched chain amino acid (BCAA) supplementation plus placebo or BCAA supplementation plus a prebiotic (7.5 gram of inulin/FOS) gut microbiome modulator daily for 12 weeks. Regular resistance exercise was prescribed to all participants. The outcomes were physical function and cognition. The trial was carried out remotely using video visits, online questionnaires, food diaries and cognitive testing, and posting of equipment and biological samples. Authors reported that the prebiotic supplement was well tolerated (although 22% reported side effects) and did result in a changed gut microbiome [e.g., increased Bifidobacterium]. There was no significant difference between prebiotic and placebo for the primary outcome of chair rise time ($\beta=0.579$; 95% CI -1.080-2.239 $p=0.494$). The Prebiotic improved cognition (first factor score versus placebo ($\beta=0.482$; 95% CI 0.141-0.823; $p=0.014$)). Authors concluded that the results demonstrate that cheap and readily available gut microbiome interventions may improve cognition in ageing population. They also concluded that the results showed the feasibility of remotely delivered trials for older people. This is a carefully designed study with multiple strengths including strong scientific premise, use of twins, well-defined outcome measures, use of appropriate data analysis methods, well-reasoned selected prebiotic dose, use of BCAA and study duration of 12 weeks. Conclusion is supported by the reported results.	Thank you for highlighting the strengths of our study – we appreciate you taking the time to review this manuscript.
I have the following questions/comments:	
1- It is started that prebiotic increased Bifidobacterium taxa. It should of course underline that increased (or decreased) in abundance f any bacteria taxa like increased Bifidobacterium is relative increased abundance	Thank you for comment, this is an important observation and has been updated throughout the manuscript to ensure this is clear.
2- The current definition of prebiotic is no longer defined by increase in	Many thanks for your comments. As cited in our introduction, we utilised the International

Bifidobacterium or lactobacillus – it is now defined by increased abundance of “anti-inflammatory” /protective bacteria taxa. Here the authors imply that inulin/FOS had prebiotic effects since Bifidobacterium was increased. This point should be stated in the discussion where the authors discussed lack of impact of prebiotic on muscle strength. Indeed, one potential explanation is that inulin/FOS might not be the appropriate prebiotic (microbiota modulator) for muscle strength in elderly. Prior studies reported that inulin/FOS might have a pro-inflammatory effect which could mitigate their microbiota modifying effects for change in muscle strength.	Scientific Association for Probiotics and Prebiotics (ISAPP) definition for a prebiotic. This consensus definition is “a substrate that is selectively utilised by host microorganisms conferring a health benefit”. https://doi.org/10.1038/nrgastro.2017.75 Bifidobacterium is widely considered to confer health benefits on the host (PMID: 27379055). It is indeed possible that inulin/FOS is not the best possible prebiotic available for influencing muscle strength, and we have added an additional paragraph to the Results Section 4.1.
3- The correlation between Bifidobacterium and improved cognition in prebiotic arm is confusing. It is reported there is a negative correlation between Bifidobacterium and improved cognition and yet prebiotic increased Bifidobacterium and improved one marker of cognition. This required explanation. Also, it appears that there is no correlation between Bifidobacterium and cognition in placebo arm . was there any correlation at baseline? If indeed Bifidobacterium plays a role in cognition, there should also be a correlation at baseline too.	Thank you for pointing out this seemingly contrasting finding. The finding seems to have been introduced by a bias in baseline abundances together with the lack of change in Actinobacteria for the placebo arm that was hard to properly control for with the non-parametric methods used so far. To ensure the validity of the findings in the manuscript, we have repeated all microbiome analyses with a linear modelling framework that also allows for compositional bias correction, which is a common challenge when analysing microbiome data (LinDA; Zhou et al. 2022, https://pubmed.ncbi.nlm.nih.gov/35421994/). The framework allows us to perform correlation analyses between microbiota features and clinical variables while adjusting for baseline abundances and other covariates. As can be seen in Figure E8 and described in Section 3.3.4, we identify Actinobacteria as positively correlated with cognition independently of twin pairing and study arm allocation.
4- It is rather surprising that the microbiota community at the end of the study was not dissimilar in twin pair when one had prebiotic and another had placebo, considering that as a group prebiotic modified microbiota community. I acknowledge, as authors stated, that the core microbiota community is established in the first 3 years of life (inheritance, vertical mother/infant transfer of microbiota and	Thank you for your comment. Indeed, we tested whether there was a dissimilarity between twin pairs at study end and included the results in our manuscript. No significant dissimilarity was found at baseline or study end point. The effect of the 12-week intervention was not sufficient to produce detectable dissimilarity between these highly similar twin pairs.

early life events) , but still environmental events in later life (e.g. antibiotics, infection, colonoscopy, stress, diet) profoundly (at least temporarily) impact microbiota community even in twins. Could lack of dissimilarity at the end of 12 weeks suggest partial recovery of prebiotic-induced microbiota community changes? This requires additional discussion.	While the lack of dissimilarity at the study end could suggest partial recovery of prebiotic changes, three time points would be needed to test this. For the PROMOTe trial we analysed the gut microbiota from faecal samples at two time points, baseline, and study end (12 weeks). As shown in Figure 15.2, there were seventeen microbiota features which differed between the prebiotic and placebo arms, highlighting that prebiotic-induced microbiota changes were identifiable at this time-point. Without a third mid-study sample, we cannot say whether there were a range of changes earlier in the intervention period, which had subsequently recovered by week 12. Future research could investigate mid-study samples to answer this research question. A greater sample size would be needed to have sufficient power to test this. We have now included this point in the study limitations.
--	---

Reviewer 3

Reviewer Comment	Response
In light of the global demographic shift towards an aging population, a deeper understanding of cognitive shifts in the elderly has become paramount for researchers in this domain. Ni Lochlainn et al. present a rigorous randomized controlled trial (RCT) evaluating the potential of gut microbiome modulation via prebiotics to enhance muscle functionality. The clinical implications and relevance of this study are articulated with precision. The utilization of twin pairs in the study design strengthens the internal validity of the trial. Notably, the remote delivery of this trial underscores a significant innovation, facilitating the inclusion of often under-represented segments of the elderly population. Nevertheless, I harbor major reservations concerning the	Thank you – we appreciate you taking the time to review this manuscript. We agree that showing remotely delivered trials are feasible in older adults is a particular strength.

statistical methodologies employed and the ensuing results.	
1. The manuscript contains a discrepancy regarding the statistical tests applied to beta diversities. Specifically, lines 744 and 795 reference the Mann-Whitney U test, yet lines 796-797 make mention of the PERMANOVA test. I would kindly request clarification on which statistical method was employed in the context of Figure E4 and E5.	Thank you for your comments. PERMANOVA tests were used for assessment of differences in gut microbiome composition between prebiotic and placebo groups at baseline and study end point (Section 3.3.2). To answer the question of whether gut microbiome composition is more similar within twin pairs than between two unrelated study participants, we used the Bray-Curtis dissimilarity of all possible participant combinations. Here, we stratified the data according to whether the two samples were twins or not for each timepoint. We then used the Mann-Whitney U (MWU) test to assess these differences as shown in Figure E4. The MWU test statistics were extracted and compared to the MWU test statistics obtained by permuting the twin/non-twin group labels (Figure E5) using a MWU test. This has been clarified in the text in Section 3.3.1 and 3.3.2. The choice of statistical test for Figure E5 has been added to the appropriate part of Section 3.3.1.
2. Although the primary focus of this study is not solely on the microbiome analysis between treatment groups, and it largely serves an exploratory analysis, it's crucial to emphasize the need for a more rigorous analysis plan, especially given the innovative nature of this study. (1) The manuscript lacks crucial details regarding the form of microbial abundance used. Was relative abundance (or proportions) employed, or were raw counts analyzed? It's a well established fact that microbiome data are inherently compositional. Directly comparing raw counts using standard statistical tests (like the t-test) would be statistically invalid. Conversely, comparisons involving relative abundance can be nuanced and pose interpretative challenges. Any change in the absolute abundance of one taxon has ramifications for the overall microbial profile. Consequently, relative abundance is	Thank you for the comments – any lack of clarity re. the use of statistical tests and abundance metrics were unintended. We have clarified the use of relative abundances throughout the manuscript and updated the Methods Section 11.4. We have taken your point on board and re-performed our microbiome analysis in line with your comments (please see below). We have now applied LinDa to all differential abundance and microbiome correlation analysis to address the compositionality of the relative abundance data. We believe this really strengthens our manuscript and thank you for your input.

often used for a global test. It would be highly advisable for the authors to review all microbiome features incorporated in this study and refine their analytical approaches accordingly.	
(2) There appears to be an absence of a comprehensive differential abundance (DA) analysis, even though Figure 2 seems to touch upon this. In the context of a mixed-effects model, I'd suggest considering more sophisticated tools like LinDA (https://cran.rproject.org/web/packages/MicrobiomeStat/index.html), ANCOM-BC2 (https://bioconductor.org/packages/release/bioc/vignettes/ANCOMBC/inst/doc/ANCOMBC2.html), MaAsLin2 (https://hub.hawaii.edu/maaslin/), or other available methods.	Thank you for these excellent suggestions for methods for differential abundance analysis. The results presented so far have been analysed using non-parametric methods, such as the Mann-Whitney U test (i.e., the Wilcoxon rank-sum test) and the Wilcoxon signed-rank test. The Wilcoxon test is the most commonly used test for microbiome abundance data. Its limited set of assumptions make it applicable to deal with the non-standard distribution of microbiome abundance data. It is acknowledged that the compositional nature of the data is not taken into account, which might lead to false discoveries. We have used LinDA to redo all differential abundance analysis (presented in Figures 2 and E6, described in Section 3.3.2) and correlation analysis between microbiome features and physical/cognitive ability (Figures E8, E9, and E10, described in Sections 3.3.3 and 3.3.4). The main findings of e.g. increased Bifidobacterium in the prebiotic group and associations between alpha-diversity and chair-stand-time were replicated but as expected, some of the findings were changed by using a linear modelling framework rather than non-parametric tests. The results, discussion, materials and methods, and all figure captions have been updated to reflect the change of statistical framework.
3. Sample size calculation needs clarifications. When I attempted to verify the sample size through manual computation, considering an SD (σ) of 0.126 and mean difference (Δ) equal to $\log_{10} 0.8$, my results were different from what's presented in the manuscript. Additionally, when I utilized the <code>`pwr.t.test`</code> function in R, the outputs were again slightly varying. On another note, assuming the mentioned	Many thanks for your comments on our manuscript, and for clarifying this. The sample size calculation was done utilising Stata, with the code used as follows:  • <code>di log10(.8)</code> • <code>power twomeans 1 `=1-.09691001' , sd(.126) power(.8)</code> This gave a sample size of 28 per group, total of 56. Allowing for 20% dropout rates, this would give a total sample size of 70.

sample size of 30 per group is accurate, by factoring in the anticipated 20% dropouts, it would necessitate recruiting 37.5 individuals per group, not 35. I might be missing some context or specific details, so I'd appreciate any clarifications on this matter.	The section of the manuscript has now been updated to ensure this is clear.
4. Reproducibility. I noticed that while the software utilized was provided, the specific code or scripts employed for the analyses were not shared. Making this available would greatly enhance the transparency and allow for more in-depth assessment and reproduction of the findings.	The codebase used for the microbiome profiling and statistical analysis is intellectual property belonging to the CRO Clinical Microbiomics. Furthermore, scripts related to the microbiome statistical analysis are part of a proprietary software package and are not meaningful as stand-alone scripts.
1. There seems to be a discrepancy between Table 3 and line 137 in the main text. The coefficients and 95% CI for the Cognition factor score in Table 3 are presented as -0.48 and [-0.81, 0.141], respectively. However, line 137 in the main text depicts all values as positive. Kindly verify and rectify for consistency.	Many thanks for your detailed review. This has now been amended to reflect the correct sign. The analysis was rerun to double check.
2. I observed that lines 137-138 in the main text do not include the coefficient, 95% CI, and pvalue for PAL. To ensure comprehensive representation, it would be helpful to have these details included. Moreover, please ensure that the signs used for these values are consistent with those in other sections of the paper.	Many thanks for your comment, this has now been included, with correct signs throughout.
3. To maintain terminological consistency, I suggest changing the title of Figure E5 from "Wilcoxon test" to "Mann-Whitney U test."	Thank you for your comment, this has been changed.
4. Regarding Figure 2, it would be beneficial to differentiate the two groups more clearly by opting for more contrasting color sets, perhaps red and blue. Additionally, the directionality of the "Change between baseline & study end" on the x-axis could be made more explicit. If I understand correctly from line 182, it should represent	Thank you for this suggestion. We have updated the figure colours and clarified the caption, where the beginning now reads: "Paired group comparison of relative abundance (a) and prevalence (p) of bacterial taxa between study arms cross-sectionally or at study end adjusted for baseline taxon abundances." Please note that analysis of bacterial abundances "at study end adjusted for baseline taxon abundances" for these purposes is

the difference calculated as "study end minus baseline".

practically equivalent to using precalculated abundance differences between study end and baseline as input for a differential abundance analysis.

REVIEWERS' COMMENTS

Reviewer #1 (Remarks to the Author):

The authors have now included four review articles to support the claim that the gut microbiota has a role in anabolic resistance. However, there is no primary evidence to support this link and it is currently a speculative mechanism. Conversely, there are numerous studies demonstrating that gut microbiota modulation (germ-free mice, antibiotic treatment) impacts muscle mass via changes in catabolic pathways (e.g. <https://pubmed.ncbi.nlm.nih.gov/31341063/> and <https://pubmed.ncbi.nlm.nih.gov/30115857/>).

I strongly disagree with the authors comment that a 75.6 kcal/d reduction in energy intake would be of no biological significance. This would have amounted to over 6000 kcal throughout the 12-week intervention that could have impacted muscle mass and strength. Was there any change in body weight in the prebiotic group? I can only find baseline body weights presented in Table 1. This reduction in energy intake is an important observation and previous studies have reported that inulin/FOS can modulate energy intake and subjective appetite. This outcome needs further comment in the Discussion, as any intervention that promotes appetite loss/anorexia in older adults would have limited utility as a strategy to maintain muscle mass.

Maltodextrin has been commonly used as a placebo vs. inulin, however the authors should acknowledge its limitations as an inert placebo. It is common when using maltodextrin as a placebo vs. a soluble fibre to provide it as an energy-matched control (e.g. <https://pubmed.ncbi.nlm.nih.gov/34170392/>), as maltodextrin would provide 4 kcal/g compared to 1.5 kcal/g for inulin. The placebo group would therefore have received an additional 1500 kcal over the 12-week supplementation period. A non-prebiotic fibre (e.g. cellulose) would have been a superior choice of placebo and overcome these limitations to the study.

Reviewer #2 (Remarks to the Author):

Authors responded to my comments (and other reviewers) satisfactorily

Reviewer #3 (Remarks to the Author):

I appreciate the authors' efforts in addressing my comments. The majority of my concerns have been appropriately resolved. However, I still have some minor inquiries related to the phrasing of the statistical analyses:

1. The LinDA method appears to be inadequately cited. While I may have overlooked it, I was unable to locate the corresponding citation in the References section.
2. In lines 202-203, the statement, "Correlation analysis between change in chair rise time and change in

microbiota features over the study intervention period..." lacks clarity. Could you specify the correlation measure utilized? I presume it to be Pearson correlation coefficients. Additionally, it would be helpful to clarify how changes in microbiota features were computed. Were raw counts used, or bias-corrected abundances from LinDA? Please provide these specifics within the Methods section for clarity.

Effect of gut microbiome modulation on muscle function and cognition: the PROMOTe randomised controlled trial: Responses to Reviewers

Reviewer 1

Reviewer Comment	Response
The authors have now included four review articles to support the claim that the gut microbiota has a role in anabolic resistance. However, there is no primary evidence to support this link and it is currently a speculative mechanism. Conversely, there are numerous studies demonstrating that gut microbiota modulation (germ-free mice, antibiotic treatment) impacts muscle mass via changes in catabolic pathways (e.g. https://pubmed.ncbi.nlm.nih.gov/31341063/ and https://pubmed.ncbi.nlm.nih.gov/30115857/).	Many thanks for your further review of our manuscript. We have now included these additional references in our introduction section and expanded on the introduction to include catabolic pathways as a postulated mechanism for the gut-muscle axis. “This highlights the role of microbiota in characterising metabolic phenotypes. Several mechanisms have been proposed for anabolic resistance, and the gut microbiome has been speculated to play a role in many of these ^{6,19-22}. Examples include protein digestion and absorption, gut barrier function, and inflammation ⁶. Further, there is evidence that the gut microbiome may influence skeletal muscle via catabolic pathways ^{23,24} “
I strongly disagree with the authors comment that a 75.6 kcal/d reduction in energy intake would be of no biological significance. This would have amounted to over 6000 kcal throughout the 12-week intervention that could have impacted muscle mass and strength. Was there any change in body weight in the prebiotic group? I can only find baseline body weights presented in Table 1. This reduction in energy intake is an important observation and previous studies have reported that inulin/FOS can modulate energy intake and subjective appetite. This outcome needs further comment in the Discussion, as any intervention that promotes appetite loss/anorexia in older adults would have limited utility as a strategy to maintain muscle mass.	Many thanks for your comments. There were no significant changes in body weight or BMI in either study arm across the intervention period. These results have now been added to Table S6. We have added the following to the results section to include this and further address the energy reduction: “The effect of this small energy intake reduction compounding over time, potentially mediated via the effect of inulin on appetite (PMIDs: 23887189; 26500686), could contribute to impacts on muscle strength, however we are cognisant of the limitations of being

	able to measure such small changes in energy intake even when using gold standard dietary recording techniques in free—living individuals. There were no significant differences in body weight (kg) or body mass index between baseline and study end in either arm (Table S6).”
Maltodextrin has been commonly used as a placebo vs. inulin, however the authors should acknowledge its limitations as an inert placebo. It is common when using maltodextrin as a placebo vs. a soluble fibre to provide it as an energy-matched control (e.g. https://pubmed.ncbi.nlm.nih.gov/34170392/), as maltodextrin would provide 4 kcal/g compared to 1.5 kcal/g for inulin. The placebo group would therefore have received an additional 1500 kcal over the 12-week supplementation period. A non-prebiotic fibre (e.g. cellulose) would have been a superior choice of placebo and overcome these limitations to the study.	Many thanks for your comments. We have now included further detail on this in the Limitations section of our manuscript. “Maltodextrin is a commonly used placebo in trials of prebiotics, however it has limitations, as it is not directly energy matched with the prebiotic. Future trials could use an energy matched placebo to overcome this.”

Reviewer 2

Reviewer Comment	Response
Authors responded to my comments (and other reviewers) satisfactorily.	Thank you - we appreciate you taking the time to review this manuscript.

Reviewer 3

Reviewer Comment	Response
I appreciate the authors' efforts in addressing my comments. The majority of my concerns have been appropriately resolved. However, I still have some minor inquiries related to the phrasing of the statistical analyses:	Thank you – we appreciate you taking the time to review this manuscript.
1. The LinDA method appears to be inadequately cited. While I may have overlooked it, I was unable to locate the corresponding citation in the References section.	Many thanks for pointing this out, the citation has been amended so that the method is clearly cited.
2. In lines 202-203, the statement, "Correlation analysis between change in	Thank you for your comments. We have added the following clarifier that states that

chair rise time and change in microbiota features over the study intervention period..." lacks clarity. Could you specify the correlation measure utilized? I presume it to be Pearson correlation coefficients. Additionally, it would be helpful to clarify how changes in microbiota features were computed. Were raw counts used, or bias-corrected abundances from LinDA? Please provide these specifics within the Methods section for clarity.

it was Pearson's correlation coefficients used.
"Correlation analysis (Pearson's correlation coefficient from models using centred log-ratio transformed abundances)-between change in chair rise time and change in microbiota features over the study intervention period..."

To compute the changes in microbiota features raw relative abundances were used and the resulting "delta values" were then centred log ratio (CLR)-transformed and used as input for the linear models. This has been added to the Methods Section 5.6: Statistical Analysis, as follows:

"To compute the changes in microbiota features raw relative abundances were used and the resulting delta values were then centred log ratio transformed and used as input for the linear models."